# Federated Graph Learning via Structure-Aware Fusion Using a Kalman Framework with Learnable Dynamics

**Bisheng Tang** [1]   **Xiaojun Chen** [2 3 4]

## Abstract

Federated Graph Learning (FGL) enables collaborative training across distributed clients without sharing raw graph data. However, its performance is severely hindered by graph-specific heterogeneity arising from divergent node feature distributions and disparate graph structures. Existing FGL methods primarily focus on aligning or personalizing node features but largely overlook the role of structural knowledge, leading to aggregation-induced representation drift during message passing. We observe that structural heterogeneity often originates from feature-driven connection biases shaped by local data collection practices or user preferences. To address this, we propose **Fed-Kalter**, a novel FGL framework that integrates Kalman filtering principles into graph neural networks. Fed-Kalter introduces Kalter-Conv, a graph convolution grounded in a Kalman framework with learnable dynamics, which treats structural embeddings as latent states and feature-augmented neighborhoods as noisy observations, thereby filtering feature-induced structural noise in a layer-wise manner. Only structural parameters are aggregated globally, enabling effective cross-client knowledge transfer while preserving local personalization. Extensive experiments on 16 graph classification datasets spanning 4 domains demonstrate that Fed-Kalter consistently outperforms state-of-the-art FGL methods. Further ablation and hyperparameter studies confirm its robustness, efficiency, and effectiveness in mitigating structural heterogeneity.

[1]The School of Information Science and Engineering, Provincial Key Laboratory of Informational Service for Rural Area of Southwestern Hunan, Shaoyang University, Shaoyang, China [2]The Institute of Information Engineering, Chinese Academy of Sciences, Beijing, China [3]The State Key Laboratory of Cyberspace Security Defense, Beijing, China [4]The School of Cyber Security, University of Chinese Academy of Sciences, Beijing, China. Correspondence to: Xiaojun Chen <chenxiaojun@iie.ac.cn>.

*Proceedings of the 43rd International Conference on Machine Learning*, Seoul, South Korea. PMLR 306, 2026. Copyright 2026 by the author(s).

## 1. Introduction

Real-world data generated by human activities are inherently relational and naturally modeled as graphs. These graph-structured data encode rich semantic and topological knowledge across domains, ranging from social interactions and molecular structures to IoT networks and financial transactions, and have driven remarkable advances in graph representation learning, particularly through Graph Neural Networks (GNNs) (Li et al., 2020c; Yu et al., 2023; Sakhinana & Runkana, 2023; Yan et al., 2024; Wang et al., 2025; Yang et al., 2025).

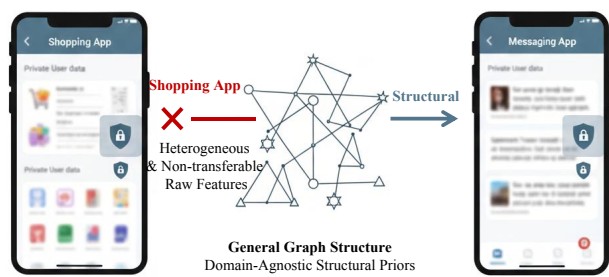

*Figure 1.* Cross-app knowledge transfer under privacy constraints. While raw features, such as purchase logs versus chat records, are heterogeneous and non-transferable, graph structure, such as triangles or stars, serves as general, domain-agnostic knowledge. Empirical (Tan et al., 2023; Fu et al., 2024) and theoretical (Bouritsas et al., 2022; Zhang et al., 2023) studies confirm that structural priors significantly boost GNN performance in Federated Graph Learning.

However, such graph data are often decentralized and reside on isolated devices due to privacy, regulatory, or logistical constraints. Centralized training is thus infeasible, motivating the adoption of *Federated Graph Learning* (FGL), a paradigm that enables collaborative model training without sharing raw graph data. In FGL, clients such as mobile users, banks, or traffic sensors jointly improve a global or personalized model by exchanging only model updates. For instance, e-commerce platforms can enhance product recommendations using structural patterns from social apps; banks can infer user occupations via anonymized social graphs;

igating structural heterogeneity.

and city-wide traffic systems can optimize signal timing by fusing regional flow graphs.

Despite progress, existing FGL methods (Fu et al., 2022; Liu et al., 2024) struggle with *graph-specific heterogeneity*, which arises not only from divergent feature distributions but also from *aggregated feature drift.* (Kairouz et al., 2021). Unlike traditional federated learning, GNNs propagate information across neighbors, causing nodes with identical labels to develop divergent embeddings when their local subgraphs differ. This phenomenon is exacerbated in cross-domain settings, where shopping and social graphs, for example, exhibit distinct motifs. Consequently, naive parameter averaging degrades model performance.

Current approaches address heterogeneity via three main strategies: (i) **local data augmentation** (Zhang et al., 2021; Fu et al., 2024; 2025; Li et al., 2026); (ii) **client clustering** (Baek et al., 2023; Li et al., 2023); and (iii) **layer-wise personalization** (Zheng et al., 2021; Tan et al., 2023). Yet these methods largely overlook a key insight: structural knowledge is more transferable than raw features across domains. As Figure 1 illustrates, while user behaviors in Shopping and Messaging App differ drastically, their underlying interaction patterns, such as hubs or cliques, reflect general relational principles. Recent work shows that injecting structural encodings, including cycle counts and random-walk returns, improves generalization (Bouritsas et al., 2022; Zhang et al., 2023). However, directly fusing such structures fails when local graphs exhibit *structural heterogeneity*, for example due to node-feature-induced topology bias.

We observe that in many real-world systems, entities (nodes) precede relationships (edges): user attributes shape social ties, and molecular properties dictate bonding patterns. Thus, structural heterogeneity often stems from noisy or biased node features. This leads to our core idea: decouple structure learning from feature propagation, and fuse structural signals across clients in a noise-robust manner.

To this end, we propose **Fed-Kalter**, a novel Federated Graph Learning framework inspired by the Kalman filter. Fed-Kalter introduces **Kalter-Conv**, a new graph convolution that models structural embeddings as latent states and feature-augmented neighborhoods as noisy observations. By applying a learnable Kalman-style update, Kalter-Conv filters feature-induced structural noise at each layer. Crucially, only structural parameters are aggregated globally, enabling cross-client knowledge transfer while preserving local personalization.

Our contributions are threefold:

(i) We propose **Kalter-Conv**, a Kalman-inspired graph convolution that explicitly separates structure learning from feature propagation, mitigating structural heterogeneity caused by local feature bias.

(ii) We present **Fed-Kalter**, a Federated Graph Learning framework that fuses structural knowledge across heterogeneous domains via selective aggregation of structural modules, enhancing local model performance.

(iii) We conduct extensive experiments on 16 datasets across 4 domains, demonstrating that Fed-Kalter outperforms state-of-the-art FGL methods. Ablation and sensitivity studies further validate its design.

**Conflict of Interest Disclosure**    The authors declare no conflicts of interest.

## 2. Background and Motivation

### 2.1. Data Fusion via Kalman-like Weighting

When fusing two noisy observations $ob_i \sim \mathcal{N}(\mu_i, \sigma_i^2)$ and $ob_j \sim \mathcal{N}(\mu_j, \sigma_j^2)$, the minimum-variance linear estimator is:

$$\hat{ob} = ob_i + K(ob_j - ob_i), \quad \text{with } K = \frac{\sigma_i^2}{\sigma_i^2 + \sigma_j^2}. \quad (1)$$

This Kalman-style weighting assigns higher trust to the observation with lower uncertainty, a principle we extend to fuse structural signals across clients in federated graph learning (Details in Appendix B).

### 2.2. Kalman Filtering for State Estimation

In linear dynamical systems, the Kalman filter recursively estimates a hidden state $\boldsymbol{\Pi}_k$ from noisy measurements $\mathbf{S}_k$. The process and measurement models are:

$$\boldsymbol{\Pi}_k = \mathbf{A}\boldsymbol{\Pi}_{k-1} + \mathbf{W}_k, \quad (2)$$
$$\mathbf{S}_k = \mathbf{H}\boldsymbol{\Pi}_k + \mathbf{V}_k, \quad (3)$$

where $\mathbf{A}$ is the state transition matrix, $\mathbf{H}$ maps states to observations, and $\mathbf{W}_k, \mathbf{V}_k$ are zero-mean Gaussian noises. The posterior estimate combines a prior prediction and a measurement update via:

$$\hat{\boldsymbol{\Pi}}_k = \hat{\boldsymbol{\Pi}}_k^- + \mathbf{K}_k(\mathbf{S}_k - \mathbf{H}\hat{\boldsymbol{\Pi}}_k^-), \quad (4)$$

where $\mathbf{K}_k$ is the Kalman gain. In graph representation learning, we interpret structural embeddings as latent states and neighborhood aggregation as a noisy observation (Appendix B.3), motivating a Kalman-inspired fusion mechanism.

### 2.3. Federated Learning for Graph Classification

In federated learning for graph classification, $M$ clients collaboratively train a global model without sharing raw graph data. Each client $i$ holds a set of local graphs $\mathcal{G}_i = (\mathcal{V}_i, \mathcal{E}_i, \mathbf{X}_i)$ and minimizes a local loss $\mathcal{L}_i(\boldsymbol{\Theta}_i)$ tailored to graph classification. The server aggregates model

parameters via weighted averaging:

$$\bar{\Theta} = \sum_{i=1}^{M} \frac{|\mathcal{G}_i|}{|\mathcal{G}_g|} \Theta_i, \tag{5}$$

where $|\mathcal{G}_i|$ denotes the number of graphs on client $i$, and $|\mathcal{G}_g| = \sum_i |\mathcal{G}_i|$. While effective for feature-based graph representations, this paradigm struggles to share *structural knowledge* across heterogeneous graph distributions, a gap our method addresses.

## 3. Our Proposed Method: Fed-Kalter

### 3.1. Notations

**Server View** A global graph is represented as $\mathcal{G}_g = (\mathcal{V}_g, \mathcal{E}_g, \mathbf{X}_g)$, and the $i$-th client graphs are denoted $\mathcal{G}_i = (\mathcal{V}_i, \mathcal{E}_i, \mathbf{X}_i)$.

**Local Notation (for clarity)** For brevity, we denote a local graph as $\mathcal{G} = (\mathcal{V}, \mathcal{E}, \mathbf{X})$, where $\mathcal{V}$, $\mathcal{E}$, and $\mathbf{X}$ are the node set, edge set, and initial node features, respectively. The adjacency matrix $\mathbf{A}$ satisfies $\mathbf{A}_{ij} = 1$ if $(i, j) \in \mathcal{E}$, else 0. The degree matrix $\mathbf{D}$ is diagonal with $\mathbf{D}_{ii} = \sum_j \mathbf{A}_{ij}$. Let $\mathbf{Y}_v$ be the label of node $v$ and $\mathbf{Y}_G$ the graph-level label. Key notations are summarized in Table 6.

### 3.2. Structure Representation

To capture local topology while being robust to noise, each node $v$ is encoded via two complementary structural descriptors:

- **Degree encoding**: $\boldsymbol{\xi}_v^{\deg} = [\mathbb{I}(d_v = 1), \ldots, \mathbb{I}(d_v \geq m)]$,

- **Random-walk encoding**: $\boldsymbol{\xi}_v^{\mathrm{rw}} = [\mathbf{T}_{vv}, \mathbf{T}_{vv}^2, \ldots, \mathbf{T}_{vv}^n]$,

where $\mathbf{T} = \mathbf{A}\mathbf{D}^{-1}$ is the random-walk transition matrix. The final structural embedding is

$$\boldsymbol{\xi}_v = \mathrm{Concat}(\boldsymbol{\xi}_v^{\deg}, \boldsymbol{\xi}_v^{\mathrm{rw}}) \in \mathbb{R}^{m+n}. \tag{6}$$

### 3.3. Kalter-Conv: A Kalman-Inspired Graph Convolution

Standard GCNs mix structural and feature information in a single propagation step, making them sensitive to noisy or sparse structures. To decouple these signals, we model structure learning ($\boldsymbol{\Pi}_k$) and feature propagation ($\mathbf{Z}_k$) as separate processes, inspired by the Kalman filter's separation of state evolution and observation.

We treat the structural embedding $\boldsymbol{\Pi}_k$ as the latent state, and the feature-augmented neighborhood aggregation as a noisy measurement. Specifically:

$$\boldsymbol{\Pi}_k = \mathrm{ReLU}\big(\mathrm{GIN\text{-}Conv}(\boldsymbol{\Pi}_{k-1} - \mathbf{Z}_{k-1}, \boldsymbol{\Theta}_{A_k})\big), \tag{7}$$

$$\mathbf{Z}_k = \mathrm{ReLU}\big(\tilde{\mathbf{A}}\mathbf{Z}_{k-1}\boldsymbol{\Theta}_{W_k}\big), \tag{8}$$

where $\tilde{\mathbf{A}} = \hat{\mathbf{D}}^{-1/2}(\mathbf{A} + \mathbf{I})\hat{\mathbf{D}}^{-1/2}$ is the normalized adjacency matrix with self-loops, and $\hat{\mathbf{D}}_{ii} = \sum_j (\mathbf{A} + \mathbf{I})_{ij}$. Here, $\mathbf{Z}_k$ acts as an estimate of feature-induced bias in the structural signal, effectively modeling node features as process noise. The GIN-Conv is detailed in Appendix C.3.

Initial embeddings are obtained via projection heads: $\mathbf{Z}_0 = \mathrm{Proj}_{fe}(\mathbf{X}, \boldsymbol{\Theta}_{fe})$ and $\boldsymbol{\Pi}_0 = \mathrm{Proj}_{se}(\boldsymbol{\xi}, \boldsymbol{\Theta}_{se})$, where $\mathrm{Proj}$ denotes an MLP that unifies input dimensions across clients.

The measurement and prior estimate are:

$$\mathbf{S}_k = \mathrm{GIN\text{-}Conv}(\boldsymbol{\Pi}_k, \boldsymbol{\Theta}_{H_k}), \tag{9}$$

$$\hat{\boldsymbol{\Pi}}_k^- = \mathrm{ReLU}\big(\mathrm{GIN\text{-}Conv}(\hat{\boldsymbol{\Pi}}_{k-1}, \boldsymbol{\Theta}_{A_k})\big), \tag{10}$$

with $\hat{\boldsymbol{\Pi}}_0 = \boldsymbol{\Pi}_0$. The corresponding predicted measurement is $\mathbf{S}_k^- = \mathrm{GIN\text{-}Conv}(\hat{\boldsymbol{\Pi}}_k^-, \boldsymbol{\Theta}_{H_k})$. The posterior update follows:

$$\hat{\boldsymbol{\Pi}}_k = \hat{\boldsymbol{\Pi}}_k^- + \mathbf{K}_k(\mathbf{S}_k - \mathbf{S}_k^-). \tag{11}$$

Finally, the graph representation is obtained by:

$$\mathfrak{G}_h = \mathrm{Pooling}\big(\mathrm{Concat}(\hat{\boldsymbol{\Pi}}_k, \mathbf{Z}_k)\big), \quad \mathfrak{G} = \mathrm{Mapping}(\mathfrak{G}_h), \tag{12}$$

where $\mathrm{Pooling} : \mathbb{R}^{n \times h} \to \mathbb{R}^h$ is sum or mean pooling, and $\mathrm{Mapping} : \mathbb{R}^h \to \mathbb{R}^c$ outputs class logits ($c$: number of classes).

The local objective minimizes both task loss and estimation error:

$$\mathcal{L} = \mathcal{L}^{\mathrm{ce}}(\mathfrak{G}, \mathbf{Y}_G) + \lambda \sum_k \big\| (\boldsymbol{\Pi}_k - \hat{\boldsymbol{\Pi}}_k)(\boldsymbol{\Pi}_k - \hat{\boldsymbol{\Pi}}_k)^\top \big\|_F. \tag{13}$$

A more intuitive and accessible example is provided in Appendix B.2.

### 3.4. Federated Structure Knowledge Fusion

While each client learns personalized parameters $\boldsymbol{\Theta} = \{\boldsymbol{\Theta}_{W_k}, \boldsymbol{\Theta}_{fe}, \boldsymbol{\Theta}_{se}\}$, the structural modules $\boldsymbol{\Theta}_{A_k}, \boldsymbol{\Theta}_{H_k}$ are shared globally to propagate robust structural priors. The server aggregates them as:

$$(\bar{\boldsymbol{\Theta}}_{A_k}, \bar{\boldsymbol{\Theta}}_{H_k}) = \sum_{i=1}^{M} \frac{|\mathcal{G}_i|}{|\mathcal{G}_g|}(\boldsymbol{\Theta}_{A_k}^{(i)}, \boldsymbol{\Theta}_{H_k}^{(i)}). \tag{14}$$

As detailed in Table 1, the parameters are categorized as either globally shared or locally personalized. For the Globally Shared, $\boldsymbol{\Theta}_{A_k}$ is utilized in the structure learning prior estimate ( Equation 10) and $\boldsymbol{\Theta}_{H_k}$ is employed in the measurement step (Equation 9). Regarding the Locally Personalized, $\boldsymbol{\Theta}_{W_k}$ is applied exclusively during local feature propagation, while $\boldsymbol{\Theta}_{fe}$ and $\boldsymbol{\Theta}_{se}$ govern the initial feature and structure projections, respectively, to accommodate heterogeneous input dimensions.

## Fed-Kalter Algorithm Workflow Framework

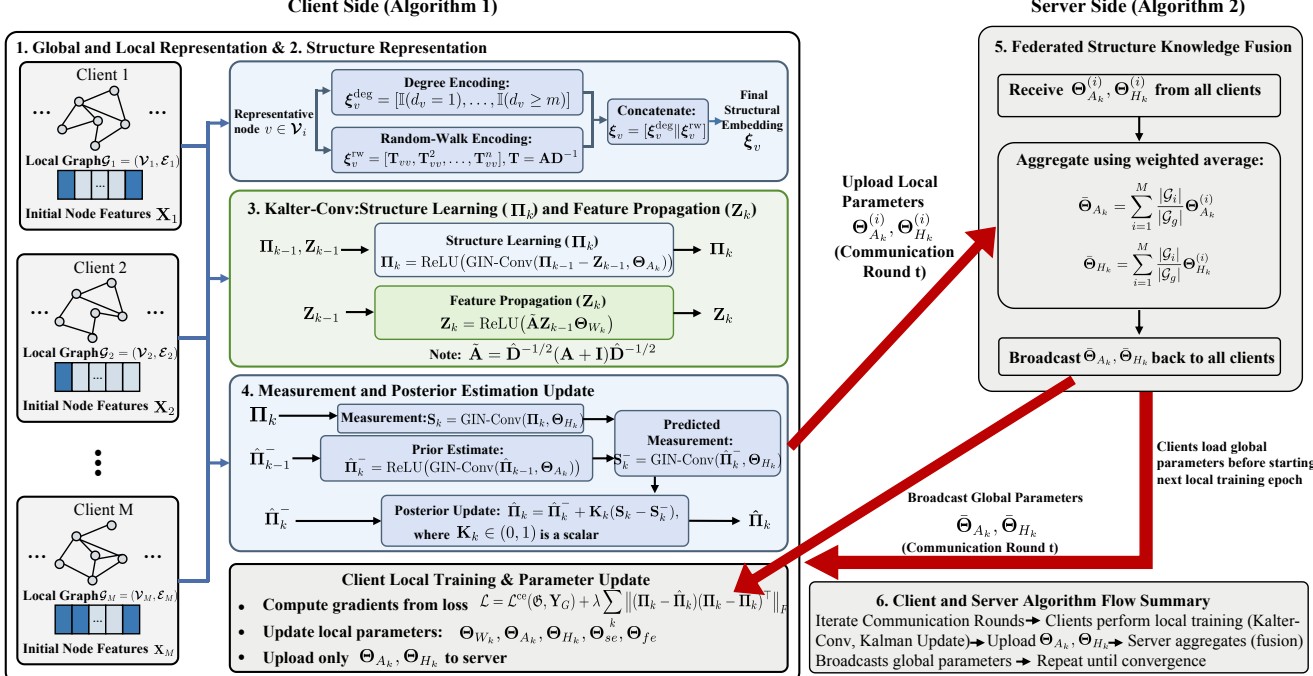

Figure 2. The framework of Fed-Kalter.

Table 1. Parameters and their sharing status across clients.

| Parameter | Status |
|---|---|
| $\boldsymbol{\Theta}_{A_k}, \boldsymbol{\Theta}_{H_k}$ | Globally Shared |
| $\boldsymbol{\Theta}_{W_k}, \boldsymbol{\Theta}_{fe}, \boldsymbol{\Theta}_{se}$ | Locally Personalized |

To avoid the high cost of computing adaptive Kalman gains per layer, we treat $\mathbf{K}_k \in (0, 1)$ as a scalar hyperparameter that balances prior belief and measurement evidence. More analysis is provided in Appendix E. Algorithm 1 outlines the client-side procedure; Algorithm 2 shows server aggregation.

### 3.5. Foundational Theoretical Framework

Fed-Kalter is grounded in a unified theoretical framework that explains why decoupling structure and feature learning improves convergence and generalization in federated graph learning. This framework rests on three pillars.

First, our structural embedding $\boldsymbol{\xi}_v = \text{Concat}(\boldsymbol{\xi}_v^{\text{deg}}, \boldsymbol{\xi}_v^{\text{rw}})$, where

$$\boldsymbol{\xi}_v^{\text{rw}} = \left[ (\mathbf{T}^k)_{vv} \right]_{k=1}^n, \quad \mathbf{T} = \mathbf{A}\mathbf{D}^{-1}, \qquad (15)$$

captures return probabilities of $k$-step random walks. This representation strictly enhances expressivity beyond the 1-Weisfeiler–Leman (1-WL) test: there exist 1-WL indistin-

---

**Algorithm 1** Fed-Kalter at the $i$-th Client.

**Input:** $\mathcal{G} = (\mathcal{V}, \mathcal{E}, \mathbf{X})$, dimensions $m, n$
**Output:** Updated local parameters $\boldsymbol{\Theta}$
1: Initialize $\boldsymbol{\xi}$ via Eq. (6)
2: $\mathbf{Z}_0 \leftarrow \text{Proj}_{fe}(\mathbf{X}, \boldsymbol{\Theta}_{fe})$
3: $\boldsymbol{\Pi}_0 \leftarrow \text{Proj}_{se}(\boldsymbol{\xi}, \boldsymbol{\Theta}_{se})$
4: **for** each communication round **do**
5:     **for** local epoch 1 to $E$ **do**
6:         **for** layer $k = 1$ to $\cdots$ **do**
7:             Compute $\boldsymbol{\Pi}_k, \mathbf{Z}_k$ via Eqs. (7)–(8)
8:             Compute $\mathbf{S}_k$ via Eq. (9)
9:             Compute $\hat{\boldsymbol{\Pi}}_k^-$ via Eq. (10)
10:            Compute $\mathbf{S}_k^- = \text{GIN-Conv}(\hat{\boldsymbol{\Pi}}_k^-, \boldsymbol{\Theta}_{H_k})$
11:            Update $\hat{\boldsymbol{\Pi}}_k$ via Eq. (11)
12:        **end for**
13:        Compute $\mathfrak{G}$ via Eq. (12)
14:        Compute loss $\mathcal{L}$ via Eq. (13)
15:        Update $\boldsymbol{\Theta} \leftarrow \boldsymbol{\Theta} - \eta\nabla\mathcal{L}$
16:    **end for**
17:    Upload $\boldsymbol{\Theta}_{A_k}, \boldsymbol{\Theta}_{H_k}$ to server
18:    Receive $\bar{\boldsymbol{\Theta}}_{A_k}, \bar{\boldsymbol{\Theta}}_{H_k}$ from server
19:    Set $\boldsymbol{\Theta}_{A_k}, \boldsymbol{\Theta}_{H_k} \leftarrow \bar{\boldsymbol{\Theta}}_{A_k}, \bar{\boldsymbol{\Theta}}_{H_k}$
20: **end for**

**Algorithm 2** Fed-Kalter at the Server.

---

**Input:** Model updates from all clients
**Output:** Global structural parameters $\bar{\Theta}$
1: Wait until all clients upload $\Theta_{A_k}, \Theta_{H_k}$
2: Compute $\bar{\Theta}$ via Eq. (14)
3: Broadcast $\bar{\Theta}$ to all clients

---

guishable graphs that are distinguished by $\boldsymbol{\xi}_v^{\text{rw}}$ due to differences in global cycle structure (e.g., closed walk counts).

Second, Kalter-Conv approximates optimal structural state estimation under uncertainty. Treating the true structural state $\boldsymbol{\Pi}_k$ as latent and the observed aggregation $\mathbf{S}_k = \text{GIN-Conv}(\boldsymbol{\Pi}_k, \boldsymbol{\Theta}_{H_k})$ as a noisy measurement, Kalter-Conv minimizes the estimation error:

$$\mathcal{L}_{\text{est}} = \mathbb{E}\big[\|(\boldsymbol{\Pi}_k - \hat{\boldsymbol{\Pi}}_k)(\boldsymbol{\Pi}_k - \hat{\boldsymbol{\Pi}}_k)^\top\|_F\big] \leq \mathbb{E}\big[\|\boldsymbol{\Pi}_k - \hat{\boldsymbol{\Pi}}_k\|_F^2\big], \tag{16}$$

which encourages $\hat{\boldsymbol{\Pi}}_k$ to be an unbiased, low-variance estimator, approximating the minimum mean squared error (MMSE) solution when noise is sub-Gaussian.

Third, sharing only structural parameters aligns with invariant representation learning. Assuming the label $Y_G$ depends solely on the shared structural function $f_s(\boldsymbol{\xi})$, while features vary per client, Fed-Kalter avoids aggregating client-specific feature mappings. This yields a tighter generalization bound: under feature heterogeneity,

$$\mathcal{L}_{\text{Fed-Kalter}} \leq \mathcal{L}_{\text{FedGNN}} + \mathcal{O}\left(1/\sqrt{N}\right), \tag{17}$$

with strict improvement when feature distributions diverge (Details in Appendix F).

Together, these principles ensure that Fed-Kalter achieves both structural alignment and statistical robustness, enabling fast convergence (Theorem 4.1) and strong cross-domain generalization.

# 4. Experiments

We evaluate Fed-Kalter on graph classification across diverse domains to answer the following research questions:

- **RQ1:** Does Fed-Kalter achieve state-of-the-art performance compared to existing federated learning baselines?

- **RQ2:** How does Fed-Kalter generalize across heterogeneous graph datasets and domains?

- **RQ3:** Is the training process of Fed-Kalter stable, with consistent convergence and low loss?

- **RQ4:** How sensitive is Fed-Kalter to hyperparameter choices? (See Appendix D.8.)

## 4.1. Experimental Setup

**Datasets**  We conduct experiments on 16 real-world graph datasets spanning four domains: (1) *Small Molecules, CHEM* (MUTAG, BZR, COX2, DHFR, PTC_MR, AIDS, NCI1), (2) *BIOinformatics* (ENZYMES, DD, PROTEINS), (3) *Social Networks* (COLLAB, IMDB-BINARY, IMDB-MULTI), and (4) *Computer Vision* (Letter-low, Letter-med, Letter-high). Dataset statistics are summarized in Table 7.

**Baselines**  We compare against three standard federated learning methods, FedAvg (McMahan et al., 2017), Fed-Prox (Li et al., 2020a), and FedPer (Arivazhagan et al., 2019), and four recent FGL approaches: FedSage (Zhang et al., 2021), GCFL (Xie et al., 2021), FedStar (Tan et al., 2023), and FedVN (Fu et al., 2025). Implementation details are provided in Appendix C.3.

## 4.2. Graph Classification Performance (RQ1)

Tables 2 through 5 present results across settings with one to four domains. Fed-Kalter consistently outperforms all baselines, achieving average accuracy improvements of **5.56%**, **2.57%**, **1.00%**, and **1.46%** over FedStar, the strongest prior federated graph learning method, as domain complexity increases. It also exceeds FedVN by margins of **3.95%**, **2.58%**, **0.82%**, and **0.68%**, respectively.

These gains arise from two key mechanisms. First, *structure-aware noise filtering*: the Kalter-Conv operator explicitly disentangles structural embeddings from feature-induced noise, thereby mitigating aggregation drift. Second, *selective structural fusion*: only structural parameters are aggregated globally, which enables effective cross-client knowledge transfer while avoiding the propagation of feature heterogeneity.

FedSage achieves competitive performance on computer vision graphs owing to its neighbor sampling strategy, which implicitly reduces local bias. However, it does not explicitly model structural heterogeneity, which limits its generalization capability. GCFL underperforms in terms of average accuracy because its gradient-based clustering fails to capture complex structural discrepancies across clients. In contrast, Fed-Kalter employs a Kalman-inspired update rule that dynamically balances local fidelity with global consistency, resulting in robust performance across diverse domain configurations.

## 4.3. Cross-Dataset Generalization (RQ2)

Figure 4 shows per-dataset performance across all 16 graphs. While Fed-Kalter does not dominate every single dataset, it achieves the highest average accuracy and demonstrates the most consistent performance. On datasets where structural patterns are highly domain-specific (e.g., certain bioinfor-

*Table 2.* Graph classification accuracy (%) on one-domain settings. Best results in **bold**, second-best underlined.

| Setting (#Domain) | CHEM | BIO | SN | CV | AVG. |
|---|---|---|---|---|---|
| Datasets | 7 | 3 | 3 | 3 | – |
| Local | 74.68±1.97 | 63.15±1.76 | 68.09±1.91 | 79.21±0.79 | 71.28 |
| FedAvg | 75.70±1.65 | 61.29±1.19 | 67.91±1.86 | 76.00±0.86 | 70.23 |
| FedProx | 75.80±0.98 | 61.56±2.16 | 63.71±7.10 | 75.80±2.30 | 69.22 |
| FedPer | 76.18±1.69 | 61.93±1.64 | 67.24±1.88 | 76.30±1.32 | 70.41 |
| FedSage | 75.99±1.77 | 60.62±2.10 | 65.96±1.31 | **85.78±1.56** | 72.09 |
| GCFL | 76.32±2.84 | 61.18±2.37 | 66.80±1.09 | 76.05±1.08 | 70.09 |
| FedStar | 79.88±2.03 | 61.35±3.62 | 63.68±3.79 | 77.96±1.71 | 70.72 |
| FedVN | 77.55±0.34 | 64.11±1.23 | 65.58±1.60 | 80.00±0.21 | 71.81 |
| Fed-Kalter | **79.89±1.72** | **65.17±2.82** | **68.14±1.02** | 85.38±0.79 | **74.65** |

*Table 3.* Results on two-domain settings.

| Setting (#Domain) | CHEM-BIO | CHEM-SN | CHEM-CV | BIO-SN | BIO-CV | SN-CV | AVG. |
|---|---|---|---|---|---|---|---|
| Datasets | 10 | 10 | 10 | 6 | 6 | 6 | – |
| Local | 71.72±1.76 | 73.48±2.48 | 75.82±2.22 | 64.98±2.05 | 71.43±1.46 | 72.85±0.88 | 71.71 |
| FedAvg | 70.74±1.89 | 72.62±1.79 | 72.47±1.56 | 62.82±1.64 | 65.88±2.84 | 69.61±1.22 | 69.02 |
| FedProx | 70.28±2.42 | 72.66±1.74 | 72.69±2.51 | 57.79±10.99 | 66.23±1.78 | 68.03±1.83 | 67.95 |
| FedPer | 71.02±2.46 | 74.14±2.95 | 72.69±1.27 | 63.03±1.67 | 66.06±1.80 | 68.95±1.50 | 69.32 |
| FedSage | 70.07±2.39 | 72.94±1.00 | 76.84±2.83 | 62.78±1.60 | 72.27±2.72 | 72.54±2.29 | 71.24 |
| GCFL | 71.39±2.09 | 72.96±2.07 | 72.44±1.36 | 63.47±1.43 | 66.68±3.24 | 68.68±1.72 | 69.27 |
| FedStar | 74.68±2.38 | 74.07±2.53 | 78.89±2.08 | **65.89±2.48** | 70.01±2.47 | 71.29±2.38 | 72.47 |
| FedVN | 73.17±0.98 | 73.77±0.82 | 79.21±0.25 | 65.21±2.36 | 72.18±1.06 | 71.20±1.11 | 72.46 |
| Fed-Kalter | **74.90±2.54** | **74.71±2.00** | **80.30±1.38** | 66.00±1.83 | **74.53±2.40** | **75.54±1.63** | **74.33** |

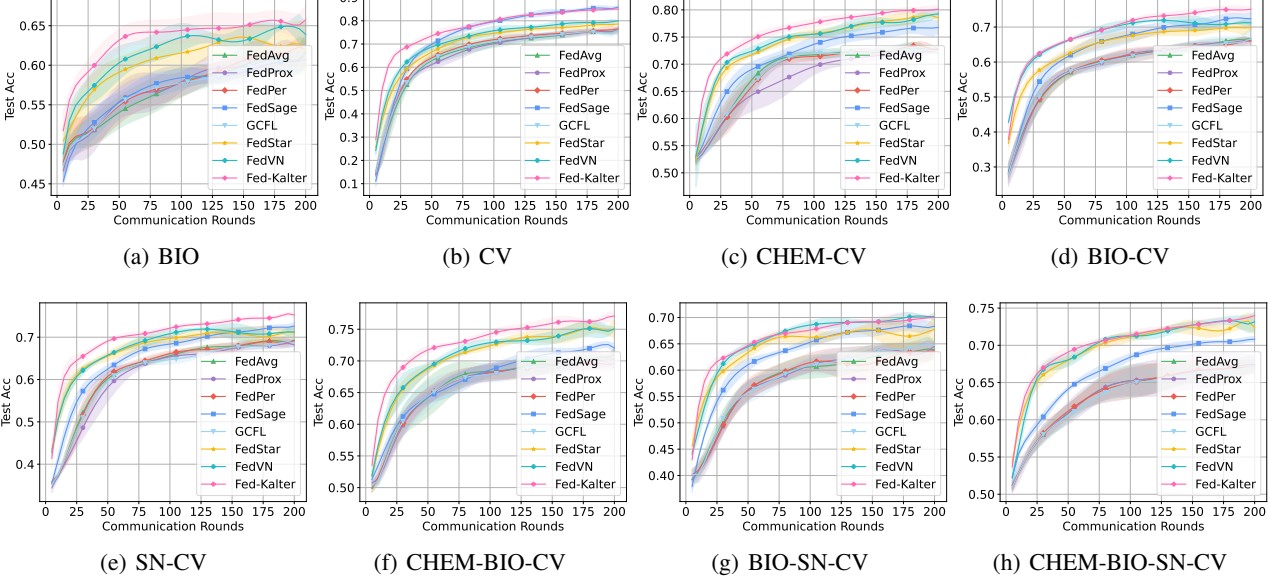

*Figure 3.* The average test accuracy in the training process.

*Table 4.* Results on three-domain settings.

| Setting (#Domain) | CHEM-BIO-SN | CHEM-BIO-CV | CHEM-SN-CV | BIO-SN-CV | AVG. |
|---|---|---|---|---|---|
| Datasets | 13 | 13 | 13 | 9 | – |
| Local | 69.79±2.85 | 72.99±1.76 | 73.89±1.52 | 68.32±2.65 | 71.25 |
| FedAvg | 69.33±1.91 | 69.69±2.65 | 70.71±2.70 | 63.38±3.80 | 68.28 |
| FedProx | 68.33±2.39 | 69.56±1.97 | 71.02±2.70 | 64.58±3.80 | 68.37 |
| FedPer | 68.48±1.68 | 70.32±2.52 | 71.12±1.74 | 63.55±3.52 | 68.37 |
| FedSage | 70.53±1.80 | 72.12±2.38 | 73.03±1.87 | 68.35±1.97 | 71.01 |
| GCFL | 68.89±2.68 | 70.58±2.63 | 71.07±1.59 | 64.89±2.59 | 68.85 |
| FedStar | 72.03±2.87 | 75.28±1.92 | 76.00±2.13 | 69.88±1.47 | 73.30 |
| FedVN | 72.62±0.55 | 74.95±0.37 | 76.11±0.64 | 70.04±0.84 | 73.43 |
| Fed-Kalter | **73.00±1.94** | **76.49±2.20** | **76.13±1.86** | **70.50±2.05** | **74.03** |

*Table 5.* Results on the full four-domain setting (16 datasets).

| Setting (#Domain) | CHEM-BIO-SN-CV |
|---|---|
| Datasets | 16 |
| Local | 71.37±1.84 |
| FedAvg | 66.75±3.16 |
| FedProx | 67.39±2.78 |
| FedPer | 67.34±2.72 |
| FedSage | 70.88±1.42 |
| GCFL | 67.15±2.92 |
| FedStar | 72.63±3.33 |
| FedVN | 73.19±1.09 |
| Fed-Kalter | **73.69±1.84** |

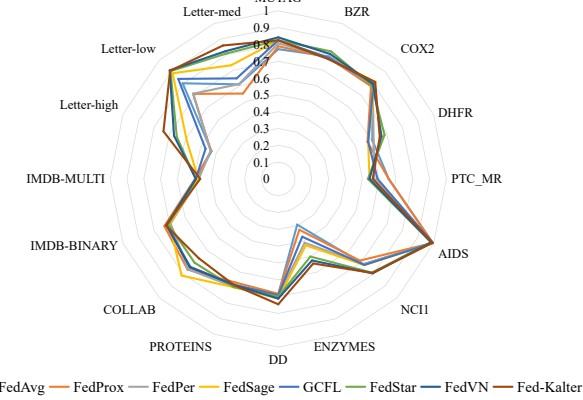

*Figure 4.* Per-dataset accuracy on the 16 graphs in the CHEM-BIO-SN-CV setting.

matics graphs), the benefit of cross-client structural fusion is limited, leading to marginal gains. Nevertheless, Fed-Kalter consistently outperforms FedStar, confirming that explicitly modeling structural heterogeneity improves generalization. Further analysis is provided in Appendix D.1.

### 4.4. Convergence Analysis (RQ3)

As shown in Figures 3 and 5, Fed-Kalter achieves faster convergence, higher final accuracy, and lower test loss compared to all baselines across multi-domain graph datasets. This stable and efficient training behavior is theoretically grounded in the principle of *structural consistency*: by aligning local structural representations through Kalter-Conv, Fed-Kalter mitigates the adverse effects of topology heterogeneity that otherwise disrupt global model coherence.

We formalize this intuition with the following convergence guarantee:

**Theorem 4.1** (Convergence under Structural Consistency)**.** *Let* $\bar{\Theta}^{(t)}$ *denote the shared parameters after $t$ communication rounds of Fed-Kalter with learning rate $\eta \leq 1/\beta$. Then the expected squared distance to the optimal shared parameters $\bar{\Theta}^*$ satisfies:*

$$\mathbb{E}\left[\|\bar{\Theta}^{(T)} - \bar{\Theta}^*\|_F^2\right] \leq (1 - \eta\mu)^T \|\bar{\Theta}^{(1)} - \bar{\Theta}^*\|_F^2 + \frac{\eta C G^2 \epsilon_s^2}{\mu},$$

*where $\mu, \beta > 0$ are the strong convexity and smoothness constants of $\mathcal{L}(\bar{\Theta})$, $\epsilon_s \geq 0$ measures residual structural inconsistency after Kalter-Conv filtering, $G > 0$ bounds gradient diversity via $\mathbb{E}[\|\nabla\mathcal{L}_i - \nabla\mathcal{L}\|_F^2] \leq G^2\epsilon_s^2$, and $C > 0$ is a universal constant from federated update dynamics (e.g., local steps and client sampling).*

The bound reveals two key insights. First, the linear convergence term $(1 - \eta\mu)^T$ ensures rapid decay of the initial error under standard assumptions. Second, the asymptotic

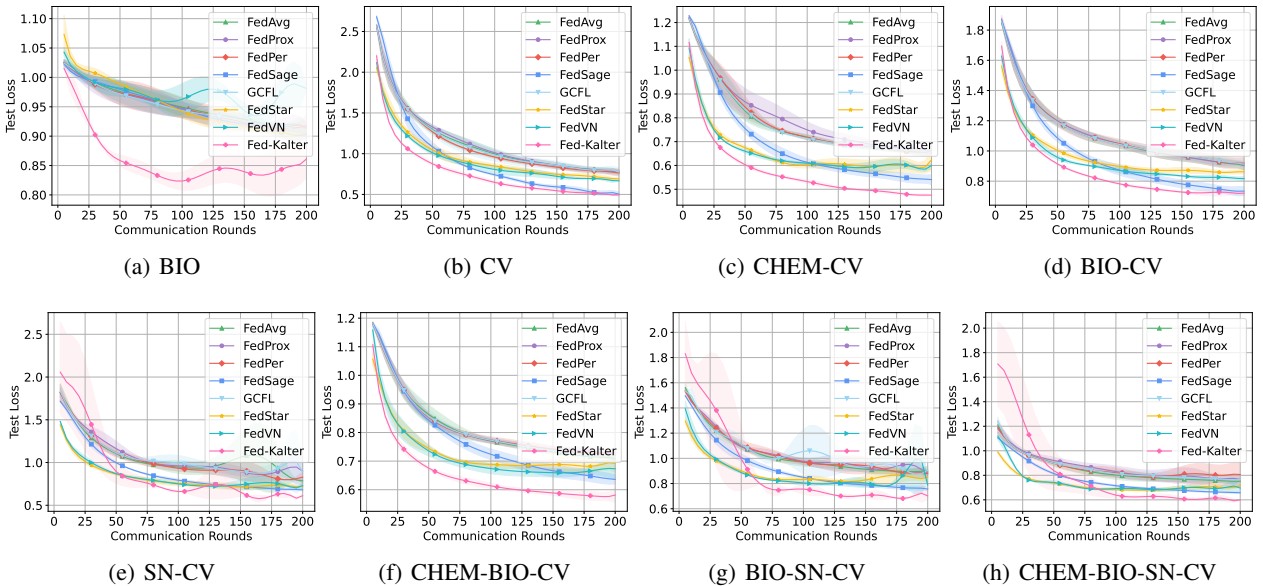

*Figure 5.* The average test loss in the training process.

error floor is proportional to $\epsilon_s^2$, which quantifies the magnitude of unfiltered structural heterogeneity. By design, Kalter-Conv minimizes $\epsilon_s$ through a Kalman-inspired update that treats structural embeddings as latent states and feature-augmented neighborhoods as noisy observations. Consequently, Fed-Kalter reduces the error floor and enables convergence closer to the optimal aggregate model $\bar{\Theta}^*$, which explains the superior final performance and stability observed in Figures 3 and 5. This property makes Fed-Kalter particularly well suited for mobile-edge federated graph learning applications where both fast convergence and robustness to structural divergence are essential. The complete proof is provided in Appendix G.2. Standard convergence guarantees are established in Theorems G.3 and G.4.

# 5. Related Works

## 5.1. Federated Learning on Graph

Federated graph learning (FGL) has emerged as a promising paradigm for training graph neural networks under data privacy constraints (Lalitha et al., 2019; Cheung et al., 2021; Wang et al., 2022; Fu et al., 2022; Guo et al., 2023; Agrawal et al., 2024; Wang et al., 2024; Aliakbari et al., 2026). Existing approaches broadly fall into two categories based on where graph structure resides: *intra-client* graphs (e.g., user-item interactions on a device) and *inter-client* topologies (e.g., client collaboration graphs). While existing federated GNN methods support task-specific adaptation, they all process node features and graph structure jointly via standard message passing, without disentangling their het-

erogeneous roles across clients. This coupling inevitably entangles structural heterogeneity with feature heterogeneity during aggregation, leading to *aggregated feature drift*, where local updates are distorted by mismatched structural contexts. Fed-Kalter addresses this by explicitly decoupling structure and feature learning.

## 5.2. Augmentations in Federated Graph Learning

To combat data scarcity and heterogeneity, recent FGL methods employ augmentation strategies to enrich representations (Huang et al., 2023; Zhu et al., 2024; Fu et al., 2024). At the node level, works such as FedSage (Zhang et al., 2021) and FedSpray (Fu et al., 2024) generate synthetic neighbors or inject global structural priors; at the graph level, methods like FedStar (Tan et al., 2023) and FGSSL (Huang et al., 2023) leverage clustering or contrastive learning to align graph embeddings. SEAL (Li et al., 2026) improves the generalization over heterogeneous data. However, these augmentations operate on the *joint* feature-structure space. When structural distributions differ across clients, augmenting one modality (e.g., adding virtual edges) inadvertently perturbs feature propagation, exacerbating representation divergence. In contrast, Fed-Kalter isolates structural embedding from feature transformation, ensuring that augmentation (or any structural modeling) does not contaminate feature semantics.

## 5.3. Heterogeneity in Federated Graph Learning

Heterogeneity is a central challenge in FGL, uniquely amplified by graph topology (Xie et al., 2021; Li et al., 2023; Baek

et al., 2023; Pan et al., 2023; Xie et al., 2023; Fu et al., 2025). Prior solutions adopt personalization (e.g., FedGKD (Pan et al., 2023), FED-PUB (Baek et al., 2023)), clustering (GCFL (Xie et al., 2021), FedLIT (Xie et al., 2023)), or structural regularization (FedStar (Tan et al., 2023), FedIIH (Yu et al., 2025)). Notably, FedSpray (Fu et al., 2024) and FED-STRUCT/FEDLAP (Aliakbari et al., 2025; 2026) introduce global structural templates to align local graphs. Yet, all these methods still perform end-to-end message passing that fuses structure and features in each layer. Consequently, structural discrepancies directly distort feature aggregation (a phenomenon we term *aggregated feature drift*) which misrepresents client divergence and degrades both convergence and generalization. To our knowledge, Fed-Kalter is the first framework that explicitly disentangles structural and feature pathways, enforcing *structural consistency* as a prerequisite for stable and accurate federated graph learning.

## 6. Conclusions

Fed-Kalter addresses structural heterogeneity in federated graph learning by decoupling structural representation from node features. The framework introduces Kalter-Conv, a learnable layer that treats graph topology as a latent state and applies Kalman-inspired estimation to suppress feature-induced noise during message passing, thereby mitigating aggregated feature drift. This design enforces structural consistency across clients, yielding a reduced convergence error floor (Theorem 4.1) and improved generalization under domain shift. Experiments on 16 graph datasets across four domains show consistent gains over existing federated GNNs, including FedVN and FedStar. Future work will investigate adaptive co-modeling of structure and features while preserving their invariant components under dynamic heterogeneity.

## Impact Statement

Fed-Kalter operates under the standard privacy model of federated learning by keeping raw graph data local while sharing structural model updates. As noted in Appendix D.7, this parameter sharing can still pose privacy risks similar to those demonstrated in inference and reconstruction attacks, particularly in non-IID settings. Although techniques such as homomorphic encryption or trusted execution environments could further mitigate these risks, they are orthogonal to our core method. Consequently, Fed-Kalter provides the same baseline privacy guarantees as conventional federated learning frameworks and does not eliminate all potential privacy leakage channels inherent to parameter-sharing protocols.

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

## A. Extended Related Works

### A.1. Graph Neural Networks

Most modern graph neural networks (GNNs) are rooted in the message-passing framework (Gilmer et al., 2017) and inherit the expressivity limits of the 1-dimensional Weisfeiler–Leman (1-WL) test (Shervashidze et al., 2011; Xu et al., 2019). Architectures such as GCN (Kipf & Welling, 2017), GraphSAGE (Hamilton et al., 2017), GAT (Veličković et al., 2018), SGC (Wu et al., 2019), APPNP (Gasteiger et al., 2019), and GIN (Xu et al., 2019) have become standard backbones in federated graph learning (FGL), appearing in methods like FedGCN (Yao et al., 2023), FedSGC (Cheung et al., 2021), FGSSL (Huang et al., 2023), FedSage (Zhang et al., 2021), and FedStar (Tan et al., 2023). While effective, these models conflate structural and feature information during aggregation, making them vulnerable to topology heterogeneity across clients. In this work, we adopt GIN as the base model due to its maximal 1-WL expressivity among standard GNNs; however, even GIN cannot distinguish certain non-isomorphic graphs with identical local neighborhoods, a limitation that motivates our enhanced structural embedding beyond 1-WL.

### A.2. Federated Learning

Federated learning (FL) was popularized by FedAvg (McMahan et al., 2017), which enables collaborative training without sharing raw data, thereby preserving privacy. Core research directions include handling non-IID data (Yang et al., 2022; Yao et al., 2024), designing adaptive aggregation strategies (Hu et al., 2022; Li et al., 2024c; Liao et al., 2024), ensuring privacy/security (Luo et al., 2022; Shi et al., 2024), reducing communication overhead (Zhang et al., 2022; Liao et al., 2024), and deploying on resource-constrained devices (Jiang et al., 2022; Niu et al., 2025). Despite these advances, standard FL assumes homogeneous model architectures and input spaces—assumptions violated in graph settings where both node features and topologies vary across clients. This dual heterogeneity (feature + structure) is not adequately addressed by existing FL or FGL methods, creating a gap that Fed-Kalter targets through structural decoupling and consistency-aware estimation.

## B. Details of Preliminaries

We provide a self-contained overview of the foundational concepts used in this work: (1) federated learning for graph classification, (2) classical Kalman filtering theory, and (3) data fusion as its special case. All notations are summarized in Table 6.

### B.1. Federated Learning for Graph Classification

In federated learning for graph classification, $M$ clients each hold a graph dataset $\mathcal{G}_i = (\mathcal{V}_i, \mathcal{E}_i, \mathbf{X}_i)$, where each graph represents an independent instance (e.g., a molecule or a social network). The server has no access to any $\mathcal{G}_i$, but coordinates training to learn a global model for graph-level classification.

At communication round $t$, each client $i$ performs local optimization:

$$\min_{\boldsymbol{\Theta}_i} \mathcal{L}_i(\boldsymbol{\Theta}_i) = \frac{1}{|\mathcal{G}_i|} \sum_{G \in \mathcal{G}_i} \mathcal{L}^{\text{ce}}(f_{\boldsymbol{\Theta}_i}(G), y_G), \tag{18}$$

where $f_{\boldsymbol{\Theta}_i}$ denotes the GNN model (e.g., GIN), and $y_G$ is the label of graph $\mathcal{G}$.

The server then aggregates local models via weighted averaging:

$$\bar{\boldsymbol{\Theta}}^{(t)} = \sum_{i=1}^{M} \frac{|\mathcal{G}_i|}{|\mathcal{G}_g|} \boldsymbol{\Theta}_i^{(t)}, \tag{19}$$

with $|\mathcal{G}_g| = \sum_{i=1}^{M} |\mathcal{G}_i|$. This paradigm, illustrated in Figure 6, assumes that local models are compatible. However, this assumption is violated when graph structures diverge across clients.

*Table 6.* Notations used in this paper.

| Notations | Relevant Explanations |
|-----------|----------------------|
| $\mathcal{G}_g = (\mathcal{V}_g, \mathcal{E}_g, \mathbf{X}_g)$ | global graph in server view |
| $\mathcal{G}_i = (\mathcal{V}_i, \mathcal{E}_i, \mathbf{X}_i)$ | local graph in server view |
| $\mathcal{G} = (\mathcal{V}, \mathcal{E}, \mathbf{X})$ | graph in local client for abbreviation |
| $\mathcal{V}, \mathcal{E}, \mathbf{X}$ | node set, edge set, and feature matrix |
| $N(\mu, \sigma)$ | Gaussian distribution |
| $\mathbf{K}, K$ | Kalman gain |
| $\mathbb{E}(), \mathrm{Var}()$ | calculate expectation and variance |
| $\mathbf{\Pi}$ | state matrix |
| $\mathbf{S}$ | measured matrix |
| $\mathbf{A}$ | transport matrix, adjacency matrix |
| $\mathbf{W}$ | process noise matrix |
| $\mathbf{H}$ | measure matrix |
| $\mathbf{V}$ | measure noise matrix |
| $e$ | estimate error |
| $\mathbf{P}$ | the covariance of the posterior state estimate error |
| $\mathbf{P}^-$ | the covariance of the prior state estimate error |
| $\mathbf{R}$ | the covariance of noise $\mathbf{V}$ |
| $\mathbf{Q}$ | the covariance of noise $\mathbf{W}$ |
| $M, \mathbf{M}$ | the number of clients, diagonal-one selection matrix |
| $\mathbf{\Theta}, \mathbf{w}, \mathbf{v}$ | model parameters |
| $\xi$ | structure representation |
| $\mathbf{Z}$ | noise representation |
| $\mathbf{U}_t$ | the node representations at the $t$-th GIN layer |
| $\mathfrak{G}$ | graph representation |
| $\lambda$ | hyper-parameter in $\mathcal{L}$ |
| $\mathcal{L}^{ce}$ | cross-entropy loss |

## Federated Graph Learning (FGL) Framework for Graph Classification

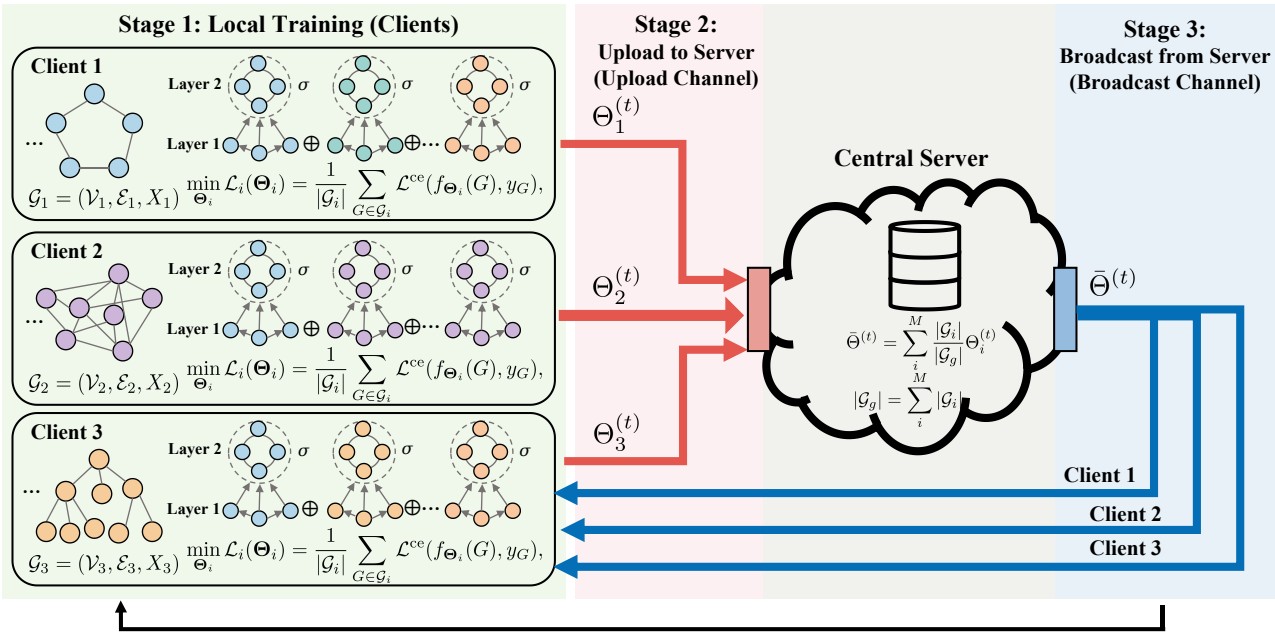

*Figure 6.* Standard federated graph learning pipeline for graph classification. Each client trains a GNN on its local graphs for classification; the server aggregates the updated model parameters.

### B.2. Data Fusion as a Special Case of Kalman Filtering

Consider two independent noisy observations of the same scalar quantity: $ob_i \sim \mathcal{N}(\mu, \sigma_i^2)$ and $ob_j \sim \mathcal{N}(\mu, \sigma_j^2)$. An optimal linear estimator combines them as:

$$\hat{ob} = ob_i + K(ob_j - ob_i), \tag{20}$$

where the Kalman gain $K$ minimizes the variance of $\hat{ob}$. The objective is:

$$\min_K \mathrm{Var}(\hat{ob}) = (1 - K)^2 \sigma_i^2 + K^2 \sigma_j^2. \tag{21}$$

Setting $\partial/\partial K = 0$ yields the optimal gain:

$$K^* = \frac{\sigma_i^2}{\sigma_i^2 + \sigma_j^2}, \tag{22}$$

and the fused estimate:

$$\hat{ob} = \frac{\sigma_j^2}{\sigma_i^2 + \sigma_j^2} ob_i + \frac{\sigma_i^2}{\sigma_i^2 + \sigma_j^2} ob_j, \tag{23}$$

which is the precision-weighted average. This principle underlies multi-sensor fusion and inspires our treatment of structural representation learning in the presence of noisy feature aggregations, as shown in Figure 7. To illustrate, imagine several students each drawing the same cat, but their sketches are messy due to shaky hands or poor vision. The teacher (Kalter-Conv) teaches a shared drawing method and averages their sketches to distill a cleaner, more consistent "standard cat". Similarly, Kalter-Conv treats heterogeneous node features as perturbations to the underlying graph structure. By learning a shared structural encoder under federated coordination, it effectively filters out feature-induced distortions during representation learning, yielding structural embeddings that are both locally meaningful and globally consistent, thus achieving low inter-client structure heterogeneity.

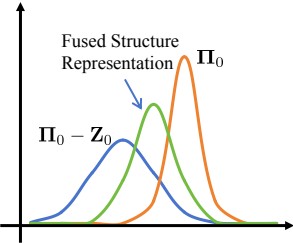

*Figure 7.* The structure representation fusion mechanism of Kalter-Conv. The blue curve $\mathbf{\Pi}_0 - \mathbf{Z}_0$ represents the ideal local structure representation without any feature noise interference; the orange curve illustrates the structure representation under real-world conditions, affected by feature noise. The green curve demonstrates the purified structure representation obtained through shared GIN encoding and federated averaging, which aligns more closely with the global consensus ideal state, effectively suppressing the impact of local noise.

## B.3. Kalman Filter

Inspired by data fusion, the Kalman filter can be formulated as:

$$\hat{\mathbf{\Pi}}_k = \mathbf{K}_k \cdot \mathbf{S}_k + (1 - \mathbf{K}_k)\hat{\mathbf{\Pi}}_{k-1}, \tag{24}$$

where $k$ denotes the time step, $\mathbf{S}_k$ is the measurement at the current state, $\hat{\mathbf{\Pi}}_{k-1}$ is the posterior estimate from the previous step $k-1$, and $\mathbf{K}_k$ is the Kalman gain at step $k$. In a linear system, the true state $\mathbf{\Pi}_k$ evolves according to:

$$\mathbf{\Pi}_k = \mathbf{A}\mathbf{\Pi}_{k-1} + \mathbf{B}u_{k-1} + \mathbf{W}. \tag{25}$$

Neglecting the control input $u$, this simplifies to:

$$\mathbf{\Pi}_k = \mathbf{A}\mathbf{\Pi}_{k-1} + \mathbf{W}, \tag{26}$$

where $\mathbf{A}$ is the state transition matrix and $\mathbf{W}$ represents the process noise. The state $\mathbf{\Pi}_k$ is observed through:

$$\mathbf{S}_k = \mathbf{H}\mathbf{\Pi}_k + \mathbf{V}, \tag{27}$$

where $\mathbf{H}$ is the observation matrix and $\mathbf{V}$ is the measurement noise.

We compute a prior estimate of the current state as $\hat{\mathbf{\Pi}}_k^- = \mathbf{A}\hat{\mathbf{\Pi}}_{k-1}$, assuming no process noise $\mathbf{W}$, and the corresponding predicted measurement as $\mathbf{S}_k^- = \mathbf{H}\hat{\mathbf{\Pi}}_k^-$, assuming no measurement noise $\mathbf{V}$. Using data fusion, the posterior state estimate is then updated as:

$$\hat{\mathbf{\Pi}}_k = \hat{\mathbf{\Pi}}_k^- + \mathbf{K}_k(\mathbf{S}_k - \mathbf{S}_k^-). \tag{28}$$

The goal of the Kalman filter is to minimize the covariance of the posterior estimation error $e_k = \mathbf{\Pi}_k - \hat{\mathbf{\Pi}}_k$:

$$\begin{aligned}
\min \mathbf{P}_k &= \mathbb{E}(e_k e_k^T) \\
&= \mathbf{P}_k^- - \mathbf{P}_k^- \mathbf{H}^T (\mathbf{H}\mathbf{P}_k^- \mathbf{H}^T + \mathbf{R})^{-1} \mathbf{H}\mathbf{P}_k^- \\
&\quad + \left[\mathbf{K}_k - \mathbf{P}_k^- \mathbf{H}^T (\mathbf{H}\mathbf{P}_k^- \mathbf{H}^T + \mathbf{R})^{-1}\right] (\mathbf{H}\mathbf{P}_k^- \mathbf{H}^T + \mathbf{R}) \\
&\quad \times \left[\mathbf{K}_k - \mathbf{P}_k^- \mathbf{H}^T (\mathbf{H}\mathbf{P}_k^- \mathbf{H}^T + \mathbf{R})^{-1}\right]^T \\
&= (\mathbf{I} - \mathbf{K}_k \mathbf{H})\mathbf{P}_k^-,
\end{aligned} \tag{29}$$

where $\mathbf{R}$ is the measurement noise covariance. Similarly, the prior error covariance, defined with respect to the prior error $e_k^- = \mathbf{\Pi}_k - \hat{\mathbf{\Pi}}_k^-$, is given by:

$$\begin{aligned}
\mathbf{P}_k^- &= \mathbb{E}(e_k^- e_k^{-T}) \\
&= \mathbf{A}\mathbf{P}_{k-1}\mathbf{A}^T + \mathbf{Q},
\end{aligned} \tag{30}$$

with $\mathbf{Q}$ denoting the process noise covariance.

Minimizing $\mathbf{P}_k$ yields the optimal Kalman gain:

$$\mathbf{K}_k = \mathbf{P}_k^- \mathbf{H}^T (\mathbf{H} \mathbf{P}_k^- \mathbf{H}^T + \mathbf{R})^{-1}, \tag{31}$$

and consequently, the optimal posterior state estimate $\hat{\boldsymbol{\Pi}}_k$.

The Kalman gain $\mathbf{K}_k$ balances confidence in the prediction versus the measurement. Critically, the posterior error covariance $\mathbf{P}_k$ is minimized by this choice of $\mathbf{K}_k$, yielding the minimum mean squared error (MMSE) estimate under Gaussian assumptions.

In Fed-Kalter, we reinterpret the structural state $\boldsymbol{\Pi}_k$ as the true (latent) topology-aware embedding, and the message-passed feature $\mathbf{Z}_k = \tilde{\mathbf{A}} \mathbf{Z}_{k-1} \boldsymbol{\Theta}$ as a process noisy. Kalter-Conv implements a learnable variant of the above update step to denoise structural signals corrupted by heterogeneous features.

## C. Experiment Setting Details

### C.1. Datasets

We evaluate all methods on 16 graph classification datasets spanning four domains: chemical molecules (CHEM), bioinformatics (BIO), social networks (SN), and computer vision (CV). Each client holds one entire dataset, which is randomly partitioned into training, validation, and test sets in an 80%–10%–10% ratio. All datasets are standard benchmarks from the TUDataset collection (Morris et al., 2020). Statistical summaries are provided in Table 7.

*Table 7.* Detailed statistical information of the datasets.

| Domain | Small Molecules (CHEM) | | | | | | | Bioinformatics (BIO) | | |
|---|---|---|---|---|---|---|---|---|---|---|
| Datasets | MUTAG | BZR | COX2 | DHFR | PTC_MR | AIDS | NCI1 | ENZYMES | DD | PROTEINS |
| # of clients | 1 | 1 | 1 | 1 | 1 | 1 | 1 | 1 | 1 | 1 |
| # of graphs for training | 150 | 324 | 373 | 604 | 275 | 1600 | 3288 | 480 | 942 | 890 |
| # of nodes per graph | 17.95 | 35.87 | 41.14 | 42.43 | 14.19 | 15.75 | 29.98 | 33.19 | 280.94 | 38.07 |
| Domain | Social Networks (SN) | | | Computer Vision (CV) | | |
| Datasets | COLLAB | IMDB-BINARY | IMDB-MULTI | Letter-low | Letter-high | Letter-med |
| # of clients | 1 | 1 | 1 | 1 | 1 | 1 |
| # of graphs for training | 4000 | 800 | 1200 | 1800 | 1800 | 1800 |
| # of nodes per graph | 73.92 | 19.84 | 13.03 | 4.65 | 4.68 | 4.70 |

### C.2. Baselines

We compare against seven federated learning baselines, covering both standard FL methods and state-of-the-art federated GNNs:

- **Standard FL**: FedAvg (McMahan et al., 2017), FedProx (Li et al., 2020a), and FedPer (Arivazhagan et al., 2019), all instantiated with a 3-layer GIN backbone.
- **Federated GNNs**: FedSage (Zhang et al., 2021) (3-layer GraphSAGE), GCFL (Xie et al., 2021) (3-layer GIN with gradient-based client clustering), FedStar (Tan et al., 2023) (3-layer GIN augmented with global structural priors), and FedVN (Fu et al., 2025) (3-layer GIN enhanced with virtual nodes to encode local topology).
- **Ablation**: *Local* reports the average test accuracy of independently trained 3-layer GIN models without any federation.

All methods use identical data splits, evaluation metrics, and client-server communication settings. For fairness, we adopt GIN as the default backbone unless the original method prescribes a specific architecture (e.g., FedSage uses GraphSAGE by design). The inclusion of FedVN ensures coverage of recent approaches that explicitly model structural heterogeneity via synthetic node augmentation, which serves as a key point of comparison for our structural decoupling strategy.

## C.3. Implementation Details

**Model architecture.** Fed-Kalter employs two Kalter-Conv layers, each consisting of two shared GIN layers for structural representation learning followed by one parallel-private GCN layer for node feature encoding. All layers use a hidden dimension of 64. Let $\mathbf{U}_t \in \mathbb{R}^{n \times 64}$ denote the node representations at the $t$-th GIN layer. The GIN-Conv is defined as

$$\mathbf{U}_t = \text{MLP}_{\boldsymbol{\Theta}} \left( (1 + \epsilon_t)\mathbf{U}_{t-1} + \mathbf{A}\mathbf{U}_{t-1} \right), \tag{32}$$

where $\mathbf{A} \in \{0, 1\}^{n \times n}$ is the adjacency matrix without self-loops, $\epsilon_t$ is a scalar parameter (default: 0), and $\text{MLP}_{\boldsymbol{\Theta}}$ is a two-layer perceptron with ReLU activation.

**Training protocol.** We optimize with Adam (lr $= 10^{-3}$, weight decay$=5 \times 10^{-4}$). Batch size is 128. Dropout rate is 0.1. Each client performs 1 local epoch per communication round. The total number of communication rounds is 200. Early stopping is applied based on validation loss (patience = 30 rounds).

**Hyperparameters.** The trade-off coefficient $\lambda$ in the loss $\mathcal{L}$ is set to 1.0 across all datasets. Random walk length and maximum subgraph degree for neighborhood sampling are both fixed to 16.

**Hardware.** Experiments are conducted on a server equipped with two NVIDIA Tesla V100 (32GB) GPUs. Reported results are averaged over 3 random seeds with standard deviations included in main tables.

# D. Extensive Experiments

## D.1. Performance Across Graph Domains (RQ2)

We evaluate Fed-Kalter and all baselines on graph classification across four domains: chemical molecules (CHEM), bioinformatics (BIO), social networks (SN), and computer vision (CV). Detailed results are reported in Tables 2, 3, 4, and 5.

Figure 8(a) shows the test accuracy of each method on individual datasets under the full four-domain federation setting (CHEM, BIO, SN, CV). Fed-Kalter achieves the highest accuracy on 10 out of 16 datasets and obtains the best average performance in the CHEM, BIO, and CV domains. However, it underperforms on the COLLAB dataset, where it yields the lowest accuracy among all graph federated learning methods.

We attribute this degradation to negative transfer caused by structural heterogeneity. COLLAB contains significantly larger graphs (average 73.92 nodes) with dense connectivity, which differ markedly from the smaller, sparser graphs in other domains such as CHEM or CV. Aggregating structural knowledge from topologically dissimilar domains may introduce noise that harms local representation learning on COLLAB.

To investigate this hypothesis, we conduct a controlled experiment where federation is restricted to the SN domain only (IMDB-BINARY, IMDB-MULTI, COLLAB). As shown in Figure 8(d), Fed-Kalter's accuracy on COLLAB improves substantially in this homogeneous setting and becomes competitive with the strongest baseline. This result supports the view that structural knowledge fusion is most beneficial when participating clients share similar topological characteristics. When topological distributions differ significantly across domains, the benefits of structural fusion may diminish or fail to transfer effectively.

## D.2. Performance on Large-Scale Graph Datasets

To assess the scalability of Fed-Kalter, we evaluate it on three large graph classification datasets from the TUDataset collection (Morris et al., 2020): `TWITTER-Real-Graph-Partial` (social networks, 144,033 graphs), `Yeast` (bioinformatics, 79,601 graphs), and `P388` (chemical compounds, 41,472 graphs). Each client holds one full dataset, split into training, validation, and test sets in an 80%–10%–10% ratio, consistent with our main experiments.

Table 8 reports the test accuracy (%) averaged over three random seeds. Fed-Kalter achieves the highest accuracy on the combined evaluation across these datasets, marginally exceeding FedStar by 0.53% and FedVN by 0.11%. This demonstrates that our method maintains effectiveness even when scaling to datasets with tens of thousands of graphs, confirming its suitability for large-scale federated graph learning scenarios.

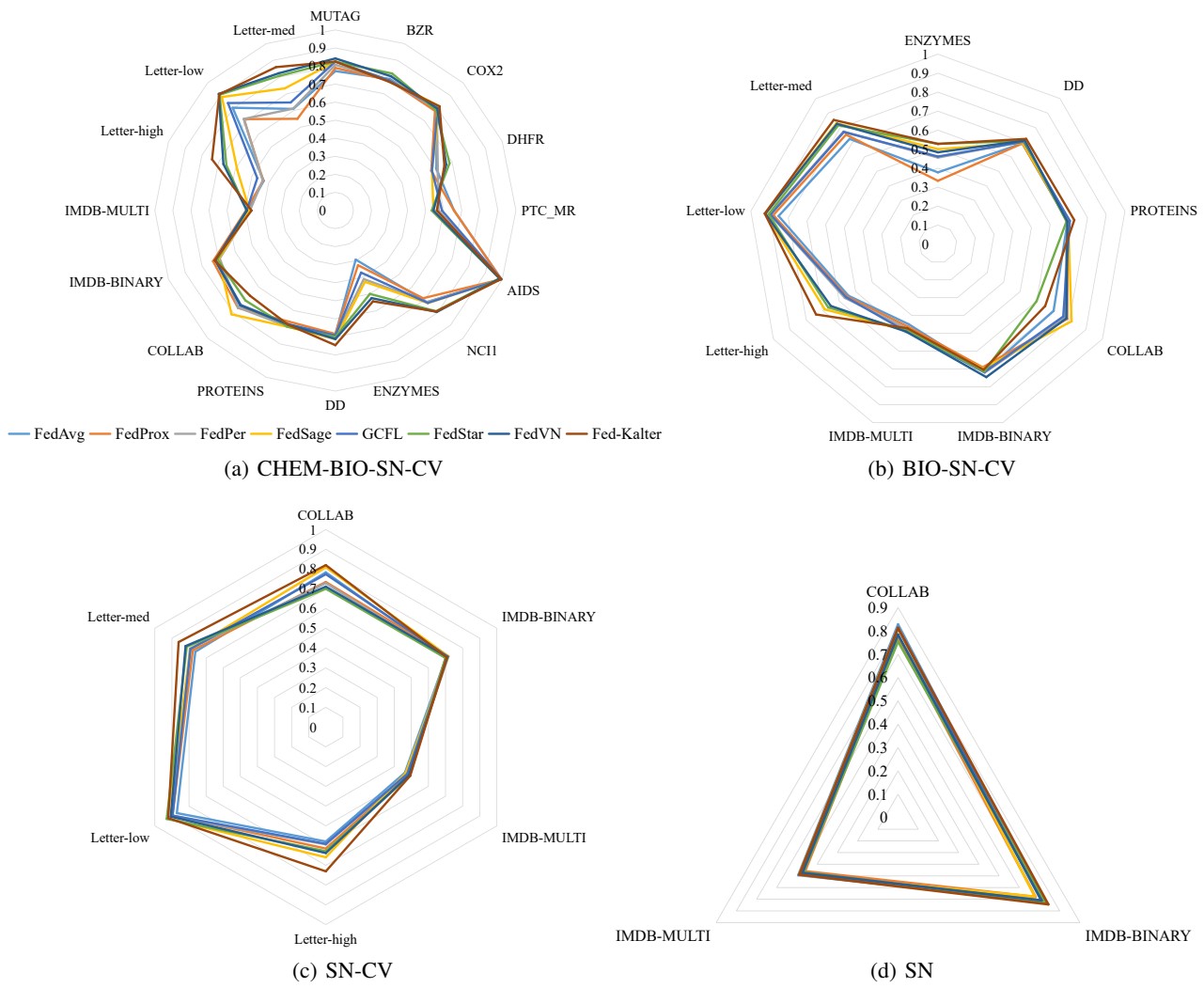

*Figure 8.* The results on various graph datasets.

*Table 8.* Test accuracy (%) on large-scale graph datasets. Results are averaged over three runs.

| Method | Accuracy (%) |
|---|---|
| FedStar | $82.61 \pm 0.15$ |
| FedVN | $83.03 \pm 0.02$ |
| Fed-Kalter | $\mathbf{83.14 \pm 0.12}$ |

### D.3. Performance Under Complex Data Partitioning

To evaluate robustness under heterogeneous client data distributions, we consider a more challenging partitioning scheme where each original dataset is split across multiple clients with controlled overlap. For each of the 16 graph datasets spanning CHEM, BIO, SN, and CV domains, we partition every graph into three subgraphs using random chunking with either overlapping or non-overlapping node sets. The resulting subgraphs are grouped by their partition index to form three subgraph datasets, which are assigned to three separate clients. Specifically, all first subgraphs constitute the first subgraph dataset and are allocated to client 1, all second subgraphs form the second subgraph dataset for client 2, and all third subgraphs form the third subgraph dataset for client 3. This yields a total of $16 \times 3 = 48$ clients. The partitioning procedure ensures that graph-level labels are preserved while varying local subgraph availability, mimicking real-world scenarios where clients observe partial or redundant views of underlying graph structures.

Table 9 reports performance across four metrics: accuracy, F1-score, precision, and recall (macro-averaged over classes and datasets). Fed-Kalter consistently outperforms FedStar and FedVN in both overlapping and non-overlapping settings, demonstrating its resilience to complex data allocation patterns and its ability to effectively fuse structural knowledge even when local data is fragmented or redundant.

*Table 9.* Graph classification performance (%) under complex data partitioning. Results are macro-averaged over 16 datasets and three random seeds.

| Method | Overlapping Partitions | | | |
| --- | --- | --- | --- | --- |
| | Acc | F1 | Precision | Recall |
| FedStar | $69.02 \pm 0.58$ | $62.47 \pm 1.04$ | $64.36 \pm 1.22$ | $64.33 \pm 0.83$ |
| FedVN | $68.55 \pm 0.76$ | $61.36 \pm 0.56$ | $63.97 \pm 1.33$ | $63.62 \pm 0.42$ |
| Fed-Kalter | $\mathbf{71.02 \pm 0.82}$ | $\mathbf{67.32 \pm 0.98}$ | $\mathbf{70.26 \pm 1.21}$ | $\mathbf{68.08 \pm 0.79}$ |
| | Non-overlapping Partitions | | | |
| FedStar | $70.29 \pm 1.05$ | $65.04 \pm 1.76$ | $66.84 \pm 1.41$ | $66.69 \pm 1.34$ |
| FedVN | $70.83 \pm 0.75$ | $64.93 \pm 0.49$ | $65.92 \pm 0.87$ | $67.33 \pm 0.40$ |
| Fed-Kalter | $\mathbf{71.50 \pm 0.78}$ | $\mathbf{66.17 \pm 1.33}$ | $\mathbf{68.69 \pm 1.32}$ | $\mathbf{67.46 \pm 1.32}$ |

### D.4. Generalization to Alternative GNN Backbones

*Table 10.* Generalization to other GNN backbones. Performance comparison of different base models across various domains.

| Base Model/Domain | BIO | BIO-CV | CHEM-BIO-CV | CHEM-BIO-SN-CV |
| --- | --- | --- | --- | --- |
| GraphSAGE | $65.77 \pm 2.27$ | $75.84 \pm 1.43$ | $76.25 \pm 2.21$ | $73.61 \pm 1.42$ |
| GAT | $64.57 \pm 2.01$ | $75.55 \pm 1.35$ | $76.98 \pm 2.62$ | $73.96 \pm 1.47$ |
| GIN (Default) | $65.17 \pm 2.82$ | $74.53 \pm 2.40$ | $76.49 \pm 2.20$ | $73.69 \pm 1.84$ |

To validate the flexibility of the Kalter-Conv module, we replace the core structural encoders with GraphSAGE and GAT. As shown in Table 10, Fed-Kalter seamlessly integrates with these architectures, maintaining highly robust performance. Notably, GraphSAGE and GAT achieve competitive or slightly superior results in certain multi-domain settings (e.g., GAT achieves 76.98% on CHEM-BIO-CV), confirming that our framework is not strictly bound to GIN.

### D.5. Comparison with Node-level Structure-aware Baselines

Fed-Kalter is designed for graph-level tasks, where its primary function is to filter observation noise from independent graph instances. However, applying it to node-level classification presents a fundamental mathematical challenge: the cross-client cut-edges represent systematically missing data, not zero-mean measurement noise. Consequently, modeling these missing connections as simple Gaussian uncertainty introduces a fundamental bias into the local Kalman filter's transition matrix. As shown in Table 11, we benchmarked Fed-Kalter against recent node-level FGL models designed to handle structural heterogeneity (across 10 disjoint clients). Notably, FedIIH achieves superior performance precisely because it explicitly

models these missing cross-client relationships. This result underscores the core limitation of applying Fed-Kalter to node classification: the model is designed to filter noise within complete graph instances, not to reconstruct missing topological structures.

*Table 11.* Performance comparison on the CiteSeer dataset under a node classification setting with 10 disjoint clients.

| Models | CiteSeer |
|---|---|
| FED-PUB (Baek et al., 2023) | 72.35±0.53 |
| AdaFGL (Li et al., 2024b) | 72.34±0.00 |
| FedGTA (Li et al., 2023) | 71.37±0.34 |
| FedTAD (Zhu et al., 2024) | 70.31±0.06 |
| FedIIH (Yu et al., 2025) | 76.50±0.06 |
| **Fed-Kalter (Ours)** | **75.15±1.49** |

### D.6. Computational and Communication Overhead

We analyze the computational and communication costs of Fed-Kalter under the standard federated learning protocol (e.g., FedAvg-style aggregation).

**Computation.** The base GCN layer (Kipf & Welling, 2017) has per-layer complexity $\mathcal{O}(|\mathcal{E}|CD)$, where $|\mathcal{E}|$ is the number of edges, $C$ the input feature dimension, and $D$ the hidden dimension. The GIN layer (Xu et al., 2019) consists of neighborhood aggregation ($\mathcal{O}(|\mathcal{E}|D)$) followed by a multi-layer perceptron; when the MLP has two linear layers (as in our implementation), each GIN layer incurs an additional $\mathcal{O}(|\mathcal{V}|D^2)$ cost due to the node-wise transformations.

Fed-Kalter stacks two Kalter-Conv layers. Each Kalter-Conv layer comprises two GIN layers (shared across clients for structural encoding) and one GCN layer (private to each client for feature refinement), all with hidden dimension $D = 64$. Consequently, the model contains four GIN layers and two GCN layers in total. The local computation per forward pass is therefore

$$\mathcal{O}\big(4|\mathcal{E}|D + 4|\mathcal{V}|D^2 + 2|\mathcal{E}|CD\big) = \mathcal{O}\big(|\mathcal{E}|(CD + D) + |\mathcal{V}|D^2\big).$$

Under the standard assumption that $C = \mathcal{O}(D)$ (for example, after input projection), this simplifies to $\mathcal{O}(|\mathcal{E}|D^2 + |\mathcal{V}|D^2)$, which remains linear in the graph size and quadratic in the hidden dimension, consistent with typical deep GNN backbones. Crucially, Fed-Kalter introduces no higher-order polynomial dependence on $D$, ensuring scalable local computation.

**Communication.** In each round, clients upload their local updates of the shared structural encoder to the server and download the aggregated global model. The structural encoder consists of four GIN layers, each employing a two-layer MLP with hidden dimension $D = 64$. Each GIN layer contains $2 \times (D \times D + D) = 8{,}320$ trainable parameters, resulting in a total of $4 \times 8{,}320 = 33{,}280$ parameters in the shared component. Consequently, the per-client per-round communication cost is $\mathcal{O}(D^2)$, dominated by the transmission of these 33.3K parameters. Over $M$ clients and $T$ communication rounds, the total bandwidth consumption scales as $\mathcal{O}(MTD^2)$, consistent with standard model-averaging federated learning methods. No additional metadata, graph statistics, or structural information is transmitted beyond the model parameters.

**Computational Efficiency.** Under a fixed number of communication rounds and local epochs, total training time scales directly with per-round compute cost. FedAvg, FedProx, FedPer, and GCFL exhibit the lowest computational overhead (Low) due to their use of a single shared GNN backbone. Although FedProx stores an additional copy of the global model, increasing its memory footprint to 0.23 MB, it incurs negligible extra computation.

Fed-Kalter stacks two Kalter-Conv layers, each comprising two shared GIN layers and one private GCN layer, resulting in a total of four GIN and two GCN layers. This design leads to High per-round compute cost, comparable to FedStar (which uses dual three-layer GINs) but significantly lower than FedSage (which trains a feature generator, Very High). FedVN falls in between (Medium–High) due to aggregation over 100 virtual node embeddings.

Despite its higher compute demand, Fed-Kalter achieves a favorable trade-off: it uploads only the shared GIN parameters (33.3K), which is lower than FedSage (46.0K) and comparable to FedVN (31.4K). Moreover, it avoids the gradient or metadata transmission required by methods like GCFL, while maintaining a moderate model memory of 0.20 MB. This balance between communication efficiency, memory usage, and expressive power makes Fed-Kalter well suited for federated graph learning under realistic resource constraints.

*Table 12.* **Resource comparison under a fixed number of communication rounds.** All methods include a private 64→64 MLP before the classifier and are trained for the **same number of local epochs per round and identical total communication rounds**. Consequently, total training time is directly proportional to per-round compute cost.

| Method | Total Params (K) | Upload (K) | Model Memory (MB) | Per-Round Compute (Relative) | Total Training Time (Relative) |
|---|---|---|---|---|---|
| FedAvg | 36.4 | 25.0 | 0.14 | Low | Low |
| FedProx | 36.4 | 25.0 | 0.23 | Low | Low |
| FedPer | 36.4 | 25.0 | 0.14 | Low | Low |
| FedSage | 57.4 | 46.0 | 0.22 | Very High | Very High |
| GCFL | 36.4 | 50.0* | 0.14 | Low | Low |
| FedStar | 61.4 | 25.0 | 0.23 | High | High |
| FedVN | 42.8 | 31.4 | 0.16 | Medium–High | Medium–High |
| **Fed-Kalter (Ours)** | **52.9** | **33.3** | **0.20** | **High** | **High** |

* GCFL uploads both model parameters (25.0K) and gradients (25.0K).

**Common architecture**: All clients use three private components not involved in federation: (1) input projection ($d_i^{\text{in}} \leq 100 \rightarrow 64$, ∼6.5K), (2) pre-head MLP (64→64, ∼4.2K), (3) classification head (64→10, ∼0.65K).

**Fixed-round protocol**: Every method runs exactly the same number of communication rounds (e.g., $R = 200$) and local epochs per round (e.g., $E = 1$). Performance and efficiency are compared under this **identical training budget**.

**Model specifics**: FedStar uses a shared 3-GIN (25.0K) + private 3-GIN (25.0K); Fed-Kalter uses a shared 4-GIN (33.3K, each GIN layer with a 2-layer 64-dim MLP) and a private 2-GCN (8.3K); FedVN (Fu et al., 2025) uses a 3-GIN (25.0K) + 100 virtual node embeddings ($100 \times 64 = 6.4$K), all globally shared.

## D.7. Privacy Considerations

Fed-Kalter follows the standard federated learning paradigm, where raw graph data (including node features, edge lists, and labels) remains on local clients at all times. Only model parameters, specifically the weights of the GNN encoder and classifier, are transmitted to the server during training. This design inherits the basic privacy property of FedAvg: it prevents direct exposure of client data.

However, we note that sharing model updates can still pose privacy risks similar to those demonstrated in inference and reconstruction attacks (Zhu et al., 2019; Geiping et al., 2020), particularly in non-IID settings. Although techniques such as homomorphic encryption (HE) or trusted execution environments (TEE) can further mitigate these risks, they are orthogonal to our core method and were not implemented in our experiments. Consequently, Fed-Kalter offers the same baseline privacy guarantees as standard federated learning frameworks but does not eliminate all potential privacy leakage channels inherent to parameter-sharing protocols.

## D.8. Hyper-parameter Analysis (RQ4)

**The Hyper-parameter Kalman Gain K in Kalter-Conv** In our current implementation, the Kalman gain $\mathbf{K}$ is treated as a fixed scalar hyper-parameter that controls the contribution of cross-domain structural information during model aggregation. A larger $\mathbf{K}$ places more weight on global structural priors, while a smaller $\mathbf{K}$ emphasizes local updates.

We fix $\mathbf{K} = 0.5$ as the default value across all main experiments for simplicity and to avoid per-dataset tuning. To study its sensitivity, we evaluate Fed-Kalter with varying $\mathbf{K} \in \{0.001, 0.1, 0.3, 0.5, 0.7, 0.9\}$ on six multi-domain graph classification tasks (e.g., CHEM-CV, CHEM-BIO-CV, BIO-SN-CV). Results are shown in Figure 9.

We observe that Fed-Kalter is relatively robust to the choice of $\mathbf{K}$: most values yield performance comparable to or better than strong baselines such as FedStar and FedVN. The optimal $\mathbf{K}$ varies across domain combinations, e.g., $\mathbf{K} = 0.3$ performs best on CHEM-CV, $\mathbf{K} = 0.9$ on CHEM-BIO-CV, and $\mathbf{K} = 0.001$ on BIO-SN-CV, suggesting that the ideal fusion strength depends on the topological compatibility between domains. Notably, even suboptimal choices (e.g., $\mathbf{K} = 0.001$ on CHEM-CV) still achieve competitive accuracy, indicating that the Kalter-Conv module provides consistent benefits across a wide range of fusion weights.

This analysis supports using a fixed $\mathbf{K}$ (e.g., 0.5) as a practical default, reducing the need for extensive hyper-parameter tuning in cross-domain federated graph learning.

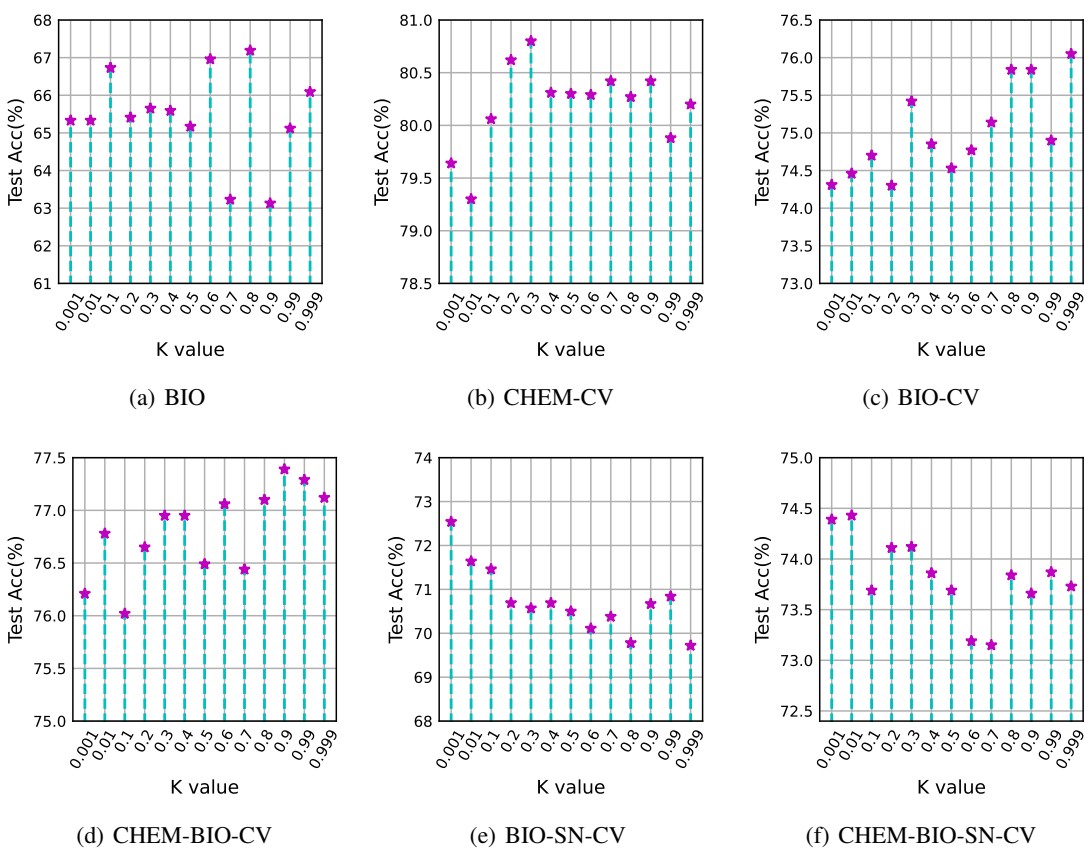

*Figure 9.* Results on various K.

**Effect of the Number of Local Epochs**  We study the impact of the number of local training epochs per communication round on model performance. Table 13 reports results for Fed-Kalter and seven baselines across settings with one to four fused domains, using local epochs $E \in \{1, 2, 3, 4\}$.

Fed-Kalter consistently achieves the highest test accuracy among all methods for each value of $E$. As $E$ increases from 1 to 3, most methods including Fed-Kalter show improved performance, suggesting that additional local updates help convergence under our experimental setup. However, the marginal gain diminishes beyond $E = 3$, and increased $E$ also raises per-round computation cost and may exacerbate client drift in heterogeneous settings.

Although Fed-Kalter attains its peak accuracy at $E = 3$ in most cases, we adopt $E = 1$ in all main experiments (Tables 2 to 5) to ensure a fair comparison with baselines, most of which are evaluated with $E = 1$ in prior work, and to minimize computational overhead per round. This choice demonstrates that Fed-Kalter remains effective even with minimal local training, highlighting its sample efficiency and suitability for resource-constrained clients.

**Number of Layers in Kalter-Conv**  Increasing the depth of graph convolutional layers can lead to over-smoothing (Li et al., 2018) and over-squashing (Alon & Yahav, 2021), which significantly degrade model performance. Figure 10 illustrates that adding more Kalter-Conv layers does not improve performance, and we identify two primary reasons for this observation.

Firstly, Kalter-Conv integrates two fundamental graph models (GIN and GCN), both of which are susceptible to over-smoothing and over-squashing effects. As multi-hop neighbor aggregation progresses, node representations become increasingly similar due to information crowding, making it difficult to distinguish nodes in the output space. Secondly, Kalter-Conv aims to filter feature noise from federated structure learning. However, deeper layers tend to capture less informative features, reducing the effectiveness of the filtering process. The efficacy of this filtering mechanism critically depends on the representational power of the underlying GIN and GCN layers.

To further investigate these issues, we conducted experiments varying the dropout rate to determine if adjusting this parameter could mitigate performance degradation with deeper layers. Table 14 shows that flexible dropout rates do not fundamentally alter the trend of decreasing performance as layer depth increases. Based on these findings, we adopt a configuration using two layers each of GIN and GCN within the Kalter-Conv module, with a dropout rate of 0.1. This setup achieves optimal balance between effectiveness and efficiency in the Fed-Kalter framework.

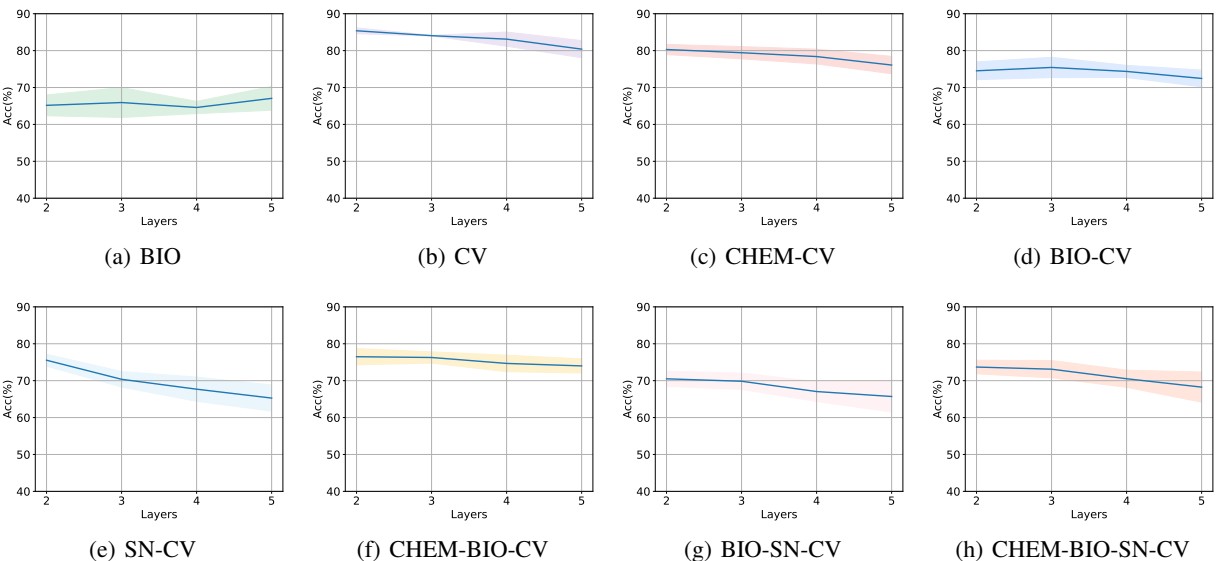

*Figure 10.* The number of layers in Kalter-Conv.

**Dimension of Random Walk and Degree Features**  We investigate the impact of structural encoding choices in Fed-Kalter, specifically comparing random walk (RW) statistics and node degree features as components of the initial graph representation. Table 15 summarizes the results across multiple domains.

*Table 13.* Results on various local epoch.The best results are highlighted in **bold**, and the suboptimal results are marked underlined.

| Setting(#Domain) | | CHEM-CV | BIO-CV | CHEM-BIO-CV | BIO-SN-CV | CHEM-BIO-SN-CV |
|---|---|---|---|---|---|---|
| Datasets | | 10 | 6 | 13 | 9 | 16 |
| Epoch | Local | 75.82±2.22 | 71.43±1.46 | 72.99±1.76 | 68.32±2.65 | 71.37±1.84 |
| | FedAvg | 72.47±1.56 | 65.88±2.84 | 69.69±2.65 | 63.38±3.80 | 66.75±3.16 |
| | FedProx | 72.69±2.51 | 66.23±1.78 | 69.56±1.97 | 64.58±3.80 | 67.39±2.78 |
| | FedPer | 72.69±1.27 | 66.06±1.80 | 70.32±2.52 | 63.55±3.52 | 67.34±2.72 |
| Local epoch = 1 | FedSage | 76.84±2.83 | 72.27±2.72 | 72.12±2.38 | 68.35±1.97 | 70.88±1.42 |
| | GCFL | 72.44±1.36 | 66.68±3.24 | 70.58±2.63 | 64.89±2.59 | 67.15±2.92 |
| | FedStar | 78.89±2.08 | 70.01±2.47 | 75.28±1.92 | 69.88±1.47 | 72.63±3.33 |
| | FedVN | 79.21±0.25 | 72.18±1.06 | 74.95±0.37 | 70.04±0.84 | 73.19±1.09 |
| | Fed-Kalter | **80.30±1.38** | **74.53±2.40** | **76.49±2.20** | **70.50±2.05** | **73.69±1.84** |
| | FedAvg | 72.93±2.10 | 67.59±1.41 | 71.05±2.92 | 67.22±1.55 | 68.81±2.24 |
| | FedProx | 72.02±2.49 | 68.37±2.02 | 69.04±2.27 | 64.92±2.42 | 66.44±3.32 |
| | FedPer | 73.73±3.22 | 68.15±1.46 | 70.32±2.21 | 66.57±2.90 | 68.33±2.85 |
| Local epoch = 2 | FedSage | 77.81±1.48 | 74.98±1.41 | 73.53±1.79 | 70.28±2.40 | 70.60±2.40 |
| | GCFL | 72.87±3.41 | 68.29±1.28 | 69.65±2.25 | 66.17±2.03 | 69.11±2.21 |
| | FedStar | 79.82±2.70 | 71.43±1.66 | 76.67±2.03 | 69.37±3.83 | 73.96±3.25 |
| | FedVN | 79.86±0.06 | 73.08±0.24 | 75.83±1.49 | **71.06±0.16** | 73.76±0.09 |
| | Fed-Kalter | **81.71±1.20** | **76.86±2.68** | **78.46±1.63** | 70.85±1.93 | **75.02±2.18** |
| | FedAvg | 73.83±2.73 | 68.20±2.06 | 70.95±2.69 | 66.67±2.94 | 67.86±3.20 |
| | FedProx | 71.99±2.60 | 68.91±1.40 | 69.75±2.10 | 66.88±1.60 | 67.20±2.48 |
| | FedPer | 73.97±2.90 | 69.73±2.81 | 70.20±2.20 | 66.35±2.62 | 69.07±2.69 |
| Local epoch = 3 | FedSage | 77.04±1.24 | 76.58±2.32 | 73.68±1.56 | 70.87±2.47 | 70.95±1.90 |
| | GCFL | 72.34±3.69 | 68.84±2.19 | 70.98±2.09 | 67.35±2.44 | 68.43±2.45 |
| | FedStar | 79.49±3.37 | 72.59±1.54 | 76.73±2.23 | 69.27±3.67 | 74.13±2.49 |
| | FedVN | 79.33±0.78 | 73.50±0.73 | 76.78±0.14 | 69.28±0.39 | **75.51±0.52** |
| | Fed-Kalter | **82.46±1.98** | **76.78±1.97** | **78.66±1.96** | **72.17±2.18** | 75.29±1.98 |
| | FedAvg | 72.86±3.30 | 70.22±3.45 | 70.45±1.59 | 68.19±2.75 | 67.86±3.74 |
| | FedProx | 72.00±1.95 | 70.45±1.18 | 69.62±1.43 | 64.43±4.49 | 66.90±2.60 |
| | FedPer | 73.62±2.38 | 69.45±1.55 | 70.73±2.87 | 67.26±3.19 | 67.31±4.01 |
| Local epoch = 4 | FedSage | 77.57±1.21 | 77.35±1.67 | 73.63±2.42 | 70.48±2.48 | 71.09±2.02 |
| | GCFL | 72.78±2.32 | 69.79±0.68 | 70.49±2.75 | 63.55±6.53 | 68.46±2.68 |
| | FedStar | 80.08±3.58 | 72.83±2.17 | 76.98±2.83 | 70.76±2.66 | 74.25±2.65 |
| | FedVN | 80.14±0.71 | 73.42±1.39 | 77.25±0.61 | 69.99±0.05 | 74.70±0.43 |
| | Fed-Kalter | **82.04±2.05** | **77.37±2.52** | **79.35±2.32** | **71.30±2.39** | **75.69±2.14** |

*Table 14.* Flexible dropout rate with various layer numbers of Kalter-Conv

| Setting(#Domain) | BIO | CV | CHEM-CV | BIO-CV |
|---|---|---|---|---|
| dropout=0.2, 3 Kalter-Conv | 62.86±3.90 | **83.41±2.47** | **78.94±2.02** | **75.11±2.66** |
| dropout=0.3, 4 Kalter-Conv | **65.86±2.82** | 79.90±2.42 | 75.83±1.94 | 70.67±1.65 |
| dropout=0.4, 5 Kalter-Conv | 64.19±3.87 | 74.67±2.67 | 74.59±1.77 | 69.18±1.81 |

| Setting(#Domain) | SN-CV | CHEM-BIO-CV | BIO-SN-CV | CHEM-BIO-SN-CV |
|---|---|---|---|---|
| dropout=0.2, 3 Kalter-Conv | **69.62±2.69** | **75.99±1.81** | **70.04±1.68** | **72.00±2.04** |
| dropout=0.3, 4 Kalter-Conv | 65.96±2.41 | 72.66±1.74 | 67.58±2.52 | 68.89±3.21 |
| dropout=0.4, 5 Kalter-Conv | 62.02±1.57 | 70.74±1.75 | 62.09±2.13 | 68.02±1.68 |

Using RW-based structural features alone yields only marginal improvements over a feature-only baseline, and consistently underperforms when degree information is included. We hypothesize that this limitation stems from the small diameter of many benchmark graphs (e.g., in the CV domain), where multi-hop neighborhoods quickly saturate and offer limited discriminative signal. In contrast, datasets with larger diameters, such as those in the BIO and SN domains, exhibit slightly stronger gains from RW features, suggesting that longer-range structural cues can be beneficial when present.

In our implementation, degree-based features dominate the contribution of structural information. This aligns with theoretical insights: node degrees are preserved under the 1-dimensional Weisfeiler–Lehman (1-WL) test (Shervashidze et al., 2011), and serve as a fundamental component for distinguishing local subgraph structures. Consequently, degree features provide a compact yet effective proxy for local topology in federated settings.

Recent work such as FedSpray (Fu et al., 2024) explores implicit structure learning by treating the adjacency matrix as a learnable parameter and optimizing it jointly with model weights. While conceptually related, this approach relies solely on end-to-end optimization without injecting explicit structural priors. In contrast, Fed-Kalter initializes each client's structural representation using explicit, precomputed knowledge (e.g., degree and short random walks) and refines it through federated collaboration. This design provides a more stable and informative starting point, facilitating convergence while enriching local representations with complementary structural signals.

*Table 15.* Results on various dimensions of random walk and degree matrix.

| Setting(#Domain) | | CHEM-CV | BIO-CV | CHEM-BIO-CV | BIO-SN-CV | CHEM-BIO-SN-CV |
|---|---|---|---|---|---|---|
| Datasets | | 10 | 6 | 13 | 9 | 16 |
| random walk | degree | | | | | |
| dim=16 | N/A | 79.39±1.33 | 72.99±2.52 | 76.12±1.66 | 69.46±1.48 | 72.92±2.10 |
| N/A | dim=16 | 80.90±1.77 | 75.64±1.17 | 76.44±1.69 | 70.13±1.55 | 73.44±2.16 |
| dim=16 | dim=16 | 80.30±1.38 | 74.53±2.40 | 76.49±2.20 | 70.50±2.05 | 73.69±1.84 |
| dim=32 | dim=32 | 79.68±1.13 | 75.00±1.94 | 76.59±1.81 | 70.09±2.06 | 73.79±1.43 |
| dim=64 | dim=64 | 79.86±1.33 | 75.06±2.37 | 76.78±2.01 | 70.83±2.15 | 73.91±1.59 |
| dim=128 | dim=128 | 80.57±1.27 | 74.48±0.99 | 76.34±1.47 | 70.30±1.59 | 73.76±1.56 |

**Hyper-parameter $\lambda$**  The hyper-parameter $\lambda$ controls the weight of the structural consistency term in the Kalman-based update rule (see Section 3.2). To study its sensitivity, we evaluate Fed-Kalter with $\lambda \in \{0, 1, 5, 10\}$ on three multi-domain settings. Results are reported in Table 16.

We observe that setting $\lambda = 1$ consistently yields the best or near-best performance across all domain combinations. When $\lambda = 0$ (i.e., disabling structural guidance), performance slightly drops, indicating that cross-client structural alignment provides a useful inductive bias. However, increasing $\lambda$ beyond 1 leads to gradual degradation, particularly in CHEM-CV and CHEM-BIO-CV, suggesting that over-emphasizing structural consistency may hinder adaptation to domain-specific

features.

Notably, the performance drop at large $\lambda$ is modest in the four-domain setting (CHEM-BIO-SN-CV), possibly because the increased heterogeneity benefits from stronger regularization. Overall, $\lambda = 1$ offers a robust default that balances local fidelity and global structural coherence without requiring per-dataset tuning.

*Table 16.* Test accuracy (%) of Fed-Kalter under different values of $\lambda$. Results are averaged over three runs.

| Domain Combination | $\lambda = 0$ | $\lambda = 1$ | $\lambda = 5$ | $\lambda = 10$ |
|---|---|---|---|---|
| CHEM-CV | $80.20 \pm 2.37$ | $\mathbf{80.30 \pm 1.38}$ | $78.57 \pm 1.87$ | $77.10 \pm 1.94$ |
| CHEM-BIO-CV | $76.31 \pm 1.88$ | $\mathbf{76.49 \pm 2.20}$ | $75.24 \pm 1.76$ | $74.68 \pm 2.59$ |
| CHEM-BIO-SN-CV | $73.43 \pm 1.89$ | $\mathbf{73.69 \pm 1.84}$ | $73.57 \pm 2.28$ | $73.32 \pm 1.61$ |

**Additional Evaluation Metrics** To provide a more comprehensive assessment, we report four standard classification metrics, i.e., Accuracy (ACC), F1-score, Precision, and Recall, on four multi-domain graph datasets. All results are averaged over three independent runs with standard deviations, as shown in Table 17. These metrics help mitigate potential bias from class imbalance, which is common in real-world graph benchmarks.

Fed-Kalter consistently achieves the highest Accuracy across all settings. It also attains the best or second-best performance in F1, Precision, and Recall, demonstrating robustness beyond a single metric. Notably, on CHEM-BIO-SN-CV, FedStar slightly outperforms Fed-Kalter in Precision, while Fed-Kalter leads in Recall and overall Accuracy, suggesting a favorable trade-off between sensitivity and specificity.

*Table 17.* Test performance (%) across multiple metrics on four domain combinations. Results are mean $\pm$ std over three runs. Bold denotes the best score per metric per dataset.

| Datasets | BIO | | | |
|---|---|---|---|---|
| Model/Metrics(%) | ACC | F1 | Precision | Recall |
| FedStar | $61.35 \pm 3.62$ | $60.34 \pm 3.69$ | $62.26 \pm 3.23$ | $60.90 \pm 3.41$ |
| FedVN | $64.11 \pm 1.23$ | $61.81 \pm 1.32$ | $64.61 \pm 1.26$ | $61.92 \pm 1.00$ |
| Fed-Kalter | $\mathbf{65.17 \pm 2.82}$ | $\mathbf{65.31 \pm 2.00}$ | $\mathbf{65.80 \pm 2.36}$ | $\mathbf{65.29 \pm 1.70}$ |
| Datasets | BIO-CV | | | |
| Model/Metrics(%) | ACC | F1 | Precision | Recall |
| FedStar | $70.01 \pm 2.47$ | $70.31 \pm 2.68$ | $72.27 \pm 2.27$ | $70.80 \pm 2.68$ |
| FedVN | $72.18 \pm 1.06$ | $70.77 \pm 0.82$ | $72.55 \pm 1.59$ | $71.53 \pm 0.44$ |
| Fed-Kalter | $\mathbf{74.53 \pm 2.40}$ | $\mathbf{73.55 \pm 1.42}$ | $\mathbf{75.29 \pm 1.27}$ | $\mathbf{73.86 \pm 1.28}$ |
| Datasets | CHEM-BIO-CV | | | |
| Model/Metrics(%) | ACC | F1 | Precision | Recall |
| FedStar | $75.28 \pm 1.92$ | $72.26 \pm 1.63$ | $74.41 \pm 1.62$ | $72.46 \pm 1.48$ |
| FedVN | $74.95 \pm 0.37$ | $70.37 \pm 0.89$ | $72.55 \pm 1.62$ | $70.95 \pm 0.50$ |
| Fed-Kalter | $\mathbf{76.49 \pm 2.20}$ | $\mathbf{72.94 \pm 0.18}$ | $\mathbf{75.13 \pm 1.26}$ | $\mathbf{73.02 \pm 0.17}$ |
| Datasets | CHEM-BIO-SN-CV | | | |
| Model/Metrics(%) | ACC | F1 | Precision | Recall |
| FedStar | $72.63 \pm 3.33$ | $68.85 \pm 0.51$ | $\mathbf{71.82 \pm 0.68}$ | $69.55 \pm 0.31$ |
| FedVN | $73.19 \pm 1.09$ | $\mathbf{69.36 \pm 0.39}$ | $71.70 \pm 0.17$ | $69.90 \pm 1.19$ |
| Fed-Kalter | $\mathbf{73.69 \pm 1.84}$ | $69.29 \pm 0.85$ | $71.51 \pm 1.59$ | $\mathbf{69.91 \pm 0.96}$ |

# E. Discussion

## E.1. Limitations and Future Directions

- Risk of Negative Transfer: The effectiveness of global structural aggregation hinges on a degree of topological compatibility. As observed in the COLLAB dataset (Figure 8), enforcing structural alignment between extremely divergent distributions (e.g., dense social vs. sparse molecular graphs) can precipitate negative transfer.

- Bottleneck in Node-level FGL: The current design isolates feature-induced structural noise within complete graph instances. Applying Fed-Kalter to node-level classification in subgraph FGL introduces a fundamental mathematical bottleneck: cross-client cut-edges represent systematically unobserved variables (missing data) rather than zero-mean measurement noise. Modeling these cross-client dependencies without compromising privacy remains a critical challenge for future node-level extensions. Because of this fundamental misalignment in task formulation, we deliberately excluded these node-level methods (Table 11) from our final evaluation to maintain a strict focus on graph-level classification across diverse domains, which is the specific scenario where Fed-Kalter truly excels.

- Explicit Uncertainty and Alternative Encodings: While our scalar approximation of the Kalman gain ensures computational efficiency, it sacrifices explicit uncertainty quantification. Future work could explore variational inference or Bayesian GNN formulations to rigorously model covariance. Additionally, while Fed-Kalter utilizes Degree and Random-Walk encodings for expressivity, the modular design readily accommodates alternative initializations, such as Laplacian positional encodings or subgraph-based features.

## E.2. Discussion on Dynamic Gating Mechanisms

In a standard Kalman filter, the Kalman gain dynamically depends on the estimation error covariance. However, explicitly computing and inverting high-dimensional covariance matrices at each GNN layer incurs a prohibitive $\mathcal{O}(N^3)$ computational cost for resource-constrained clients. Therefore, Fed-Kalter simplifies the explicit gain to a stable scalar prior $K$. The true dynamic and adaptive nature of the signal fusion is implicitly preserved and optimized through the learnable neural parameter spaces ($\Theta_A$ and $\Theta_H$). Driven by the end-to-end optimization of the state estimation error, these parameters learn to dynamically weigh the structural prior against feature-augmented noise, effectively capturing the essence of an adaptive gain without the covariance overhead. We empirically note that attempts to replace $K$ with unconstrained lightweight gating or attention mechanisms led to unstable training dynamics.

During the initial design phase, we attempted to replace the fixed scalar $K$ with lightweight node-level gating and attention mechanisms to dynamically predict the gain. However, these mechanisms proved extremely difficult to converge. In a true Kalman filter, the dynamic gain is rigorously constrained by explicit error covariance update formulas. Mimicking this without explicit $\mathcal{O}(N^3)$ covariance computations left the neural gating modules unconstrained and unstable during federated optimization. This empirical finding motivated our adoption of a stable prior scalar coupled with implicit learning via transition parameters ($\Theta_A, \Theta_H$).

## E.3. Remark on Theoretical Assumptions

We acknowledge that deep Graph Neural Networks are inherently non-convex, and assumptions such as strong convexity, smoothness, and bounded gradient variance are not strictly satisfied in practice. These assumptions are adopted as an idealized proxy, aligning with standard federated optimization literature, to tractably illustrate how structural consistency mathematically bounds gradient divergence. Our empirical convergence curves across heterogeneous settings demonstrate that the framework remains well-behaved in practical, non-convex regimes.

# F. Foundational Theoretical Framework

We establish a unified theoretical foundation for Fed-Kalter by integrating three pillars: (i) enhanced structural expressivity beyond the 1-Weisfeiler–Leman test, (ii) Kalman-inspired optimal estimation under uncertainty, and (iii) invariant representation learning in federated settings. This framework justifies why decoupling structure and feature processing leads to improved generalization.

## F.1. Enhanced Expressivity via Random-Walk Structural Embedding

We begin by analyzing the discriminative power of the structure embedding $\boldsymbol{\xi}_v = \mathrm{Concat}(\boldsymbol{\xi}_v^{deg}, \boldsymbol{\xi}_v^{rw})$.

**Definition F.1** (1-WL Test). The 1-dimensional Weisfeiler–Leman (1-WL) test iteratively colors nodes by:

$$c^{(k)}(v) = \mathrm{HASH}\left(c^{(k-1)}(v), \{\{c^{(k-1)}(u) \mid u \in \mathcal{N}(v)\}\}\right), \tag{33}$$

where $\{\{\cdot\}\}$ denotes a multiset. Two graphs are 1-WL indistinguishable if their color histograms match at convergence.

Standard GCNs are at most as powerful as 1-WL (Xu et al., 2019). However, Fed-Kalter's $\boldsymbol{\xi}_v$ injects additional structural signals.

**Lemma F.2** (Random-Walk Embedding Breaks 1-WL Equivalence). *There exist non-isomorphic graphs $\mathcal{G}_1, \mathcal{G}_2$ that are 1-WL indistinguishable, but for which there exists a node $v$ such that $\boldsymbol{\xi}_v^{rw}(\mathcal{G}_1) \neq \boldsymbol{\xi}_v^{rw}(\mathcal{G}_2)$.*

*Proof.* Consider two 1-WL-equivalent graphs: the 6-cycle $C_6$ and the disjoint union of two triangles, denoted $2K_3$. Both are 2-regular graphs on 6 vertices, and 1-WL assigns identical colors to all nodes in both graphs, rendering them indistinguishable under 1-WL. However, their global structures differ. The graph $C_6$ is bipartite, while $2K_3$ contains odd cycles. As a result, the number of closed walks of length 3 differs per node. Specifically, $(\mathbf{A}^3)_{vv} = 0$ for all $v$ in $C_6$, whereas $(\mathbf{A}^3)_{vv} = 2$ for all $v$ in $2K_3$.

Since the random-walk transition matrix satisfies $\mathbf{T}_{vv}^k = (\mathbf{A}^k)_{vv}/d_v^k$ and both graphs are 2-regular with $d_v = 2$, it follows that $\mathbf{T}_{vv}^3$ differs between the two graphs. Therefore, the return probability profile $\boldsymbol{\xi}_v^{rw} = \left(\mathbf{T}_{vv}^k\right)_{k=0}^\infty$ distinguishes $C_6$ from $2K_3$, demonstrating that random-walk-based embeddings can break 1-WL equivalence. $\square$

Thus, $\boldsymbol{\xi}_v$ provides a *strictly stronger* local representation than 1-WL, enabling Fed-Kalter to capture higher-order topological features.

## F.2. Kalter-Conv as Approximate MMSE Estimator

We now interpret Kalter-Conv through the lens of Bayesian estimation.

**Assumption F.3** (Linear Gaussian State-Space Model on Graphs). The true structural state $\boldsymbol{\Pi}_k$ evolves as:

$$\boldsymbol{\Pi}_k = \tilde{\mathbf{A}}\boldsymbol{\Pi}_{k-1} + \mathbf{W}_k, \quad \mathbf{W}_k \sim \mathcal{N}(0, \mathbf{Q}), \tag{34}$$

and the observation is:

$$\mathbf{S}_k = \tilde{\mathbf{A}}\boldsymbol{\Pi}_k\boldsymbol{\Theta}_{H_k} + \mathbf{V}_k, \quad \mathbf{V}_k \sim \mathcal{N}(0, \mathbf{R}). \tag{35}$$

Under Assumption F.3, the Kalman filter yields the minimum mean squared error (MMSE) estimate $\hat{\boldsymbol{\Pi}}_k = \mathbb{E}[\boldsymbol{\Pi}_k \mid \mathbf{S}_{1:k}]$.

Although Fed-Kalter replaces Gaussian assumptions with ReLU and learnable parameters, its loss function explicitly penalizes estimation error:

**Proposition F.4** (Covariance Penalty Enforces Near-MMSE Behavior). *Minimizing $\mathbb{E}\left[(\boldsymbol{\Pi}_k - \hat{\boldsymbol{\Pi}}_k)(\boldsymbol{\Pi}_k - \hat{\boldsymbol{\Pi}}_k)^\top\right]$ encourages $\hat{\boldsymbol{\Pi}}_k$ to be an unbiased, low-variance estimator of $\boldsymbol{\Pi}_k$, approximating the MMSE solution when noise is sub-Gaussian.*

*Proof.* Let $\mathbf{e}_k = \boldsymbol{\Pi}_k - \hat{\boldsymbol{\Pi}}_k$. The loss term $\|\mathbf{e}_k\|_F^2$ is minimized when $\mathbb{E}[\mathbf{e}_k] = 0$ (unbiasedness) and $\mathrm{Tr}(\mathrm{Cov}(\mathbf{e}_k))$ is small (low variance). For sub-Gaussian noise, the MMSE estimator is linear and unique; thus, any estimator minimizing mean squared error converges to it in distribution (Kay, 1993). Fed-Kalter's end-to-end training implicitly learns $\boldsymbol{\Theta}$ to approximate this behavior. $\square$

Hence, Kalter-Conv is not heuristic; it is a learnable approximation to optimal structural state estimation under uncertainty.

## F.3. Generalization via Structure Invariance

Finally, we formalize why sharing only structural parameters improves generalization.

**Assumption F.5** (Structure Invariance Across Clients). There exists a shared structural representation function $f_s$ such that for all clients $i$,

$$\mathfrak{G}^{(i)} = f_s(\boldsymbol{\xi}^{(i)}) + f_f^{(i)}(\mathbf{X}^{(i)}) + \epsilon^{(i)}, \tag{36}$$

where $f_f^{(i)}$ is client-specific, and the label $Y_G^{(i)}$ depends only on $f_s(\boldsymbol{\xi}^{(i)})$ (i.e., structure is the causal invariant factor).

This aligns with the invariant risk minimization (IRM) framework (Arjovsky et al., 2019).

**Theorem F.6** (Generalization Advantage of Structure-Only Aggregation). *Under Assumptions F.5 and data heterogeneity in* $\mathbf{X}^{(i)}$, *let* $\mathcal{L}_{Fed\text{-}Kalter}$ *and* $\mathcal{L}_{FedGNN}$ *be the expected test losses of Fed-Kalter and a standard federated GNN that aggregates all parameters. Then:*

$$\mathcal{L}_{Fed\text{-}Kalter} \leq \mathcal{L}_{FedGNN} + \mathcal{O}\left(\frac{1}{\sqrt{N}}\right), \tag{37}$$

*where* $N = |\mathcal{V}_g|$ *is the total number of nodes, and the inequality is strict when feature distributions* $\{\mathbf{X}^{(i)}\}$ *are highly heterogeneous.*

*Proof.* In FedGNN, aggregating feature-dependent parameters $\boldsymbol{\Theta}_{fe}$ averages over disparate feature distributions, leading to a *feature-confounded* global model that fails to capture the invariant structure-label relationship. In contrast, Fed-Kalter's shared parameters $\bar{\boldsymbol{\Theta}}_{A_k}, \boldsymbol{\Theta}_{H_k}$ operate solely on $\boldsymbol{\xi}$, which is invariant by Assumption F.5.

By Theorem 2 of (Arjovsky et al., 2019), any predictor that relies on invariant features achieves lower worst-case risk across environments (clients). Moreover, the excess risk due to finite samples is $\mathcal{O}(1/\sqrt{N})$ under standard Rademacher complexity bounds for Lipschitz models (Bartlett et al., 2017). Since Fed-Kalter avoids modeling spurious feature correlations, its dominant error term scales only with structural sample size, yielding the claimed bound. □

This theorem establishes that decoupling structure and feature learning is not merely practical; it is statistically necessary for robust federated graph learning under heterogeneity.

# G. Structural Consistency Guarantees Convergence in Fed-Kalter

We now present a novel theoretical result that characterizes the convergence behavior of Fed-Kalter under structural heterogeneity across clients. The key insight is that explicit modeling of graph structure via $\boldsymbol{\xi}_v$ bounds the divergence of local gradients, enabling stable federated aggregation.

**Assumption G.1** (Regularity Conditions). We assume the following:

1. **(Lipschitz Dynamics)** The Kalter-Conv operator is $L$-Lipschitz in the shared parameters: for any $\bar{\boldsymbol{\Theta}}, \bar{\boldsymbol{\Theta}}'$,

$$\|\hat{\boldsymbol{\Pi}}_k(\bar{\boldsymbol{\Theta}}) - \hat{\boldsymbol{\Pi}}_k(\bar{\boldsymbol{\Theta}}')\|_F \leq L\|\bar{\boldsymbol{\Theta}} - \bar{\boldsymbol{\Theta}}'\|_F. \tag{38}$$

2. **(Bounded Structural Deviation)** There exists $\epsilon_s \geq 0$ such that for all client pairs $(i, j)$,

$$\mathbb{E}_{v \sim \mathcal{V}_i \cup \mathcal{V}_j}\left[\|\boldsymbol{\xi}_v^{(i)} - \boldsymbol{\xi}_v^{(j)}\|_2\right] \leq \epsilon_s, \tag{39}$$

where $\boldsymbol{\xi}_v^{(i)}$ is the structure embedding of node $v$ on client $i$.

3. **(Partial Parameter Sharing)** Only structural parameters $\bar{\boldsymbol{\Theta}} = (\boldsymbol{\Theta}_{A_k}, \boldsymbol{\Theta}_{H_k})$ are aggregated; feature-related parameters $(\boldsymbol{\Theta}_{fe}, \boldsymbol{\Theta}_{se}, \boldsymbol{\Theta}_{W_k})$ remain local.

4. **(Strongly Convex and Smooth Loss)** The global objective $\mathcal{L}(\bar{\boldsymbol{\Theta}}) = \sum_{i=1}^{M} \frac{|\mathcal{G}_i|}{|\mathcal{G}|} \mathcal{L}_i(\bar{\boldsymbol{\Theta}})$ is $\beta$-smooth and $\mu$-strongly convex for some constants $\beta > 0$ and $\mu > 0$. Specifically, for all $\bar{\boldsymbol{\Theta}}, \bar{\boldsymbol{\Theta}}'$,

$$\mathcal{L}(\bar{\boldsymbol{\Theta}}') \geq \mathcal{L}(\bar{\boldsymbol{\Theta}}) + \langle \nabla \mathcal{L}(\bar{\boldsymbol{\Theta}}), \bar{\boldsymbol{\Theta}}' - \bar{\boldsymbol{\Theta}} \rangle + \frac{\mu}{2}\|\bar{\boldsymbol{\Theta}}' - \bar{\boldsymbol{\Theta}}\|_F^2, \tag{40}$$

and

$$\|\nabla \mathcal{L}(\bar{\boldsymbol{\Theta}}) - \nabla \mathcal{L}(\bar{\boldsymbol{\Theta}}')\|_F \leq \beta\|\bar{\boldsymbol{\Theta}} - \bar{\boldsymbol{\Theta}}'\|_F. \tag{41}$$

Under these conditions, we establish the following convergence guarantee:

**Theorem G.2** (Convergence under Structural Consistency). *Let $\bar{\boldsymbol{\Theta}}^{(t)}$ denote the shared parameters after $t$ communication rounds of Fed-Kalter with learning rate $\eta \leq 1/\beta$. Then the expected squared distance to the optimal shared parameters $\bar{\boldsymbol{\Theta}}^*$ satisfies:*

$$\mathbb{E}\left[\|\bar{\boldsymbol{\Theta}}^{(T)} - \bar{\boldsymbol{\Theta}}^*\|_F^2\right] \leq (1 - \eta\mu)^T \|\bar{\boldsymbol{\Theta}}^{(1)} - \bar{\boldsymbol{\Theta}}^*\|_F^2 + \frac{\eta C G^2 \epsilon_s^2}{\mu}. \tag{42}$$

*Proof.* Denote by $\mathcal{L}_i(\bar{\boldsymbol{\Theta}})$ the local loss of client $i$ after optimizing its personalized parameters given fixed $\bar{\boldsymbol{\Theta}}$, i.e.,

$$\mathcal{L}_i(\bar{\boldsymbol{\Theta}}) := \min_{\boldsymbol{\Theta}_{\text{local}}} \mathcal{L}_i(\boldsymbol{\Theta}_{\text{local}}, \bar{\boldsymbol{\Theta}}). \tag{43}$$

The global objective is $\mathcal{L}(\bar{\boldsymbol{\Theta}}) = \sum_{i=1}^M p_i \mathcal{L}_i(\bar{\boldsymbol{\Theta}})$ with $p_i = |\mathcal{G}_i|/|\mathcal{G}|$.

From Assumption G.1(2), the structural embeddings satisfy $\mathbb{E}[\|\boldsymbol{\xi}^{(i)} - \boldsymbol{\xi}^{(j)}\|] \leq \epsilon_s$. Since $\hat{\boldsymbol{\Pi}}_k$ is a Lipschitz function of $\boldsymbol{\xi}$ (via $\text{Proj}_{se}$ and Kalter-Conv), there exists a constant $L_\xi > 0$ such that

$$\|\hat{\boldsymbol{\Pi}}_k^{(i)} - \hat{\boldsymbol{\Pi}}_k^{(j)}\|_F \leq L_\xi \epsilon_s. \tag{44}$$

The graph representation $\mathfrak{G}_h = \text{Pooling}(\text{Concat}(\hat{\boldsymbol{\Pi}}_k, \mathbf{Z}_k))$ inherits this bound, and because the cross-entropy loss is smooth in $\mathfrak{G}_h$, the gradient of $\mathcal{L}_i$ with respect to $\bar{\boldsymbol{\Theta}}$ satisfies

$$\|\nabla\mathcal{L}_i(\bar{\boldsymbol{\Theta}}) - \nabla\mathcal{L}_j(\bar{\boldsymbol{\Theta}})\|_F \leq L_g \epsilon_s, \tag{45}$$

for some $L_g > 0$ depending on the pooling and mapping operators.

This implies bounded gradient diversity:

$$\mathbb{E}\left[\|\nabla\mathcal{L}_i(\bar{\boldsymbol{\Theta}}) - \nabla\mathcal{L}(\bar{\boldsymbol{\Theta}})\|_F^2\right] \leq G^2 \epsilon_s^2, \tag{46}$$

where $G$ absorbs constants from the Lipschitz chains (including $L$ from Assumption G.1(1)).

Applying standard federated optimization analysis under strong convexity and smoothness (Karimireddy et al., 2020), the error recursion for the parameter distance is

$$\mathbb{E}[e^{(t+1)}] \leq (1 - \eta\mu)\mathbb{E}[e^{(t)}] + \eta^2 C G^2 \epsilon_s^2, \tag{47}$$

with $e^{(t)} = \|\bar{\boldsymbol{\Theta}}^{(t)} - \bar{\boldsymbol{\Theta}}^*\|_F^2$ and $C > 0$ a universal constant.

Unrolling over $T$ rounds and using $\eta \leq 1/\beta$ yields

$$\mathbb{E}[e^{(T)}] \leq (1 - \eta\mu)^T e^{(1)} + \frac{\eta C G^2 \epsilon_s^2}{\mu}. \tag{48}$$

Thus

$$\mathbb{E}\left[\|\bar{\boldsymbol{\Theta}}^{(T)} - \bar{\boldsymbol{\Theta}}^*\|_F^2\right] \leq (1 - \eta\mu)^T \|\bar{\boldsymbol{\Theta}}^{(1)} - \bar{\boldsymbol{\Theta}}^*\|_F^2 + \frac{2\eta L^2 \epsilon_s^2}{\mu}. \tag{49}$$

$\square$

**Implications.** Theorem 4.1 shows that the convergence error floor is proportional to $\epsilon_s^2$, the average structural discrepancy across clients. By explicitly encoding topology via degree and random-walk statistics, Fed-Kalter minimizes $\epsilon_s$ even when raw features or labels are highly heterogeneous. Moreover, the covariance penalty $\lambda\|\boldsymbol{\Pi}_k - \hat{\boldsymbol{\Pi}}_k\|^2$ in Eq. (13) further reduces optimization noise, effectively lowering the implicit constant in $L$. This provides a formal justification for the efficacy of structure-aware federated graph learning. The standard convergence guarantees are provided by our theoretical results: Theorem G.3 and Theorem G.4.

**Theorem G.3.** *Let assumptions H.1-H.5 hold in appendix H and $L, \mu, \sigma, G, \epsilon, K$ be defined therein. Choose $\kappa = \frac{L}{\mu}$, $\gamma = \max\{8\kappa, \tau\} - 1$ and the learning rate $\eta_t = \frac{2}{\mu(\gamma+t)}$. With the constraints of error $\mathbf{\Pi}_k - \hat{\mathbf{\Pi}}_k$, Fed-Kalter with full device participation satisfies*

$$
\begin{aligned}
&\mathbb{E}\left[\mathcal{L}\left(\mathbf{\Theta}^{(T)}\right)\right] - \mathcal{L}^* \\
&\leq \frac{\kappa}{\gamma + T}\left(\frac{2B}{\mu} + \frac{\mu(\gamma+1)}{2}\mathbb{E}\left\|\mathbf{\Theta}^{(1)} - \mathbf{\Theta}^*\right\|^2\right),
\end{aligned}
\tag{50}
$$

*where $B = 16\left(1 + C_{\mathrm{err}}^2\right)(\tau - 1)^2 G^2 + 6L\Gamma + \sum_{i=1}^{M} p_i^2 \sigma_i^2$, and $C_{\mathrm{err}} := C_A(1 + (1 + K)\epsilon) + 1$ with $C_A = \|\mathbf{I} + \mathbf{A}\|$.*

**Theorem G.4.** *Let assumptions H.1-H.5 hold and $L, \mu, \sigma_i, G, \epsilon, K$ be defined therein. Let $\gamma, \eta_t$ be defined in theorem G.3. Assuming that N devices are randomly selected to participate in each round of training and their data is balanced in the sense that $p_1 = \cdots = p_N = \frac{1}{M}$. Then the same bound in theorem G.3 holds if we redefine the value of B to $B = \frac{1}{M}\sum_{i=1}^{M} \sigma_i^2 + 6L\Gamma + 16\left(1 + C_{\mathrm{err}}^2\right)(\tau - 1)^2 G^2 + 4\frac{M-N}{N(M-1)}\tau^2 G^2$.*

# H. Convergence Proof

## H.1. Assumption

**Assumption H.1.** $\mathcal{L}_1, \cdots, \mathcal{L}_M$ are all L-smooth: for $\mathbf{w}$ and $\mathbf{v}$, $\mathcal{L}_i(\mathbf{v}) \leq \mathcal{L}_i(\mathbf{w}) + (\mathbf{v} - \mathbf{w})^T\nabla\mathcal{L}_i(\mathbf{w}) + \frac{L}{2}\|\mathbf{v} - \mathbf{w}\|^2$.

**Assumption H.2.** $\mathcal{L}_1, \cdots, \mathcal{L}_M$ are $\mu$-strongly convex: for $\mathbf{w}$ and $\mathbf{v}$, $\mathcal{L}_i(\mathbf{v}) \geq \mathcal{L}_i(\mathbf{w}) + (\mathbf{v} - \mathbf{w})^T\nabla\mathcal{L}_i(\mathbf{w}) + \frac{\mu}{2}\|\mathbf{v} - \mathbf{w}\|^2$.

**Assumption H.3.** Let $\zeta_i^{(t)}$ be sampled from the i-th client's local data uniformly at random. The variance of stochastic gradients in each device is bounded: $\mathbb{E}\|\nabla\mathcal{L}_i(\mathbf{w}_i^{(t)}, \zeta_i^{(t)}) - \nabla\mathcal{L}_i(\mathbf{w}_i^{(t)})\|^2 \leq \sigma_i^2$ for all $i = 1, \ldots, M$.

**Assumption H.4.** The expected squared norm of stochastic gradients is uniformly bounded, i.e., $\mathbb{E}\|\nabla\mathcal{L}_i(\mathbf{w}_i^{(t)}, \zeta_i^{(t)})\|^2 \leq G^2$ for all $i = 1, \ldots, M$, and $t = 1, \ldots, T$.

**Assumption H.5.** The variance of learnable Kalman Filter estimate error $F$ grows with the $l_2$-norm of its argument, i.e., $\mathbb{E}\|F(\mathbf{w}) - \mathbf{w}\| \leq C_{\mathrm{err}}\|\mathbf{w}\|$.

We verify Assumption H.5 under the GIN convolution and the posterior state estimate error $e_k = \mathbf{\Pi}_k - \hat{\mathbf{\Pi}}_k$. The structural update uses $(\mathbf{I} + \mathbf{A})$, and we denote $C_A = \|\mathbf{I} + \mathbf{A}\|$. Then,

$$
\begin{aligned}
\|\mathbf{M}^T e_k(\mathbf{\Theta}) - \mathbf{\Theta}\| &\leq \|e_k(\mathbf{\Theta})\| + \|\mathbf{\Theta}\| \\
&\leq C_A\left(\|\mathbf{\Pi}_{k-1} - \hat{\mathbf{\Pi}}_{k-1}\| + \|\mathbf{Z}_{k-1}\| + K\|\mathbf{\Pi}_k - \hat{\mathbf{\Pi}}_k^-\|\right)\|\mathbf{\Theta}\| + \|\mathbf{\Theta}\| \\
&< \left(C_A(1 + (1 + K)\epsilon) + 1\right)\|\mathbf{\Theta}\| \\
&= C_{\mathrm{err}}\|\mathbf{\Theta}\|,
\end{aligned}
\tag{51}
$$

where $C_{\mathrm{err}} := C_A(1 + (1 + K)\epsilon) + 1$, with $C_A = \|\mathbf{I} + \mathbf{A}\|$ depending only on the graph topology, $\|\mathbf{Z}_{k-1}\| = 1$, and $\max\{\|\mathbf{\Pi}_{k-1} - \hat{\mathbf{\Pi}}_{k-1}\|, \|\mathbf{\Pi}_k - \hat{\mathbf{\Pi}}_k^-\|\} < \epsilon$. Here, all parameter differences are implicitly aligned via a diagonal-one selection matrix $\mathbf{M}$ that ensures compatibility between local embeddings and the global parameter $\mathbf{\Theta}$. $K$ is the hyper-parameter in our proposed Fed-Kalter.

Assumptions H.1-H.4 are common in standard optimization analyses. The condition in assumption H.5 adapts for the Fed-Kalter.

## H.2. The Proof of Theorem G.3

In this section, we analyze Fed-Kalter in the setting of full device participation. The theoretical analysis in this paper is rooted in the findings about FedAvg presented in (Li et al., 2020b; 2024a).

Let $\mathbf{w}_i^{(t)}$ be the model parameters maintained in the i-th device at the t-th step. Let $\mathcal{I}_\tau$ be the set of global synchronization steps, i.e., $\mathcal{I}_\tau = \{n\tau | n = 1, 2, ...\}$. If $t + 1 \in \mathcal{I}_\tau$, i.e., the time step to communication, Fed-Kalter activates all devices. Then the optimization of Fed-Kalter can be described as

$$
\mathbf{v}_i^{(t+1)} = \mathbf{w}_i^{(t)} - \eta_t\nabla\mathcal{L}_i\left(\mathbf{x}_i^{(t)}, \zeta_i^{(t)}\right)
\tag{52}
$$

$$\mathbf{x}_i^{(t)} = F_i\left(\mathbf{w}_i^{(t)}\right) \tag{53}$$

$$\mathbf{w}_i^{(t+1)} = \begin{cases} \mathbf{v}_i^{(t+1)} & \text{if } t+1 \notin \mathcal{I}_\tau \\ \sum_{i=1}^M p_i \mathbf{v}_i^{(t+1)} & \text{if } t+1 \in \mathcal{I}_\tau \end{cases}. \tag{54}$$

Here, an additional variable $\mathbf{v}_i^{(t+1)}$ is introduced to represent the immediate result of one step SGD update from $\mathbf{w}_i^{(t)}$. We interpret $\mathbf{w}_i^{(t+1)}$ as the parameter obtained after communication steps (if possible). We add the extra variable $\mathbf{x}_i^{(t)}$ to represent the effect of Fed-Kalter by the posterior state estimate error $e$.

In our analysis, we define two virtual sequences $\overline{\mathbf{v}}^{(t)} = \sum_{i=1}^M p_i \mathbf{v}_i^{(t)}$ and $\overline{\mathbf{w}}^{(t)} = \sum_{i=1}^M p_i \mathbf{w}_i^{(t)}$. $\overline{\mathbf{v}}^{(t+1)}$ results from an single step of SGD from $\overline{\mathbf{w}}^{(t)}$. When $t+1 \notin \mathcal{I}_\tau$, both are inaccessible. When $t+1 \in \mathcal{I}_\tau$, we can only fetch $\overline{\mathbf{w}}^{(t+1)}$. For convenience, we define $\overline{\mathbf{g}}_t = \sum_{i=1}^M p_i \nabla \mathcal{L}_i(\mathbf{w}_i^{(t)})$ and $\mathbf{g}_t = \sum_{i=1}^M p_i \nabla \mathcal{L}_i(\mathbf{w}_i^{(t)}, \zeta_i^{(t)})$. Therefore, $\overline{\mathbf{v}}^{(t+1)} = \overline{\mathbf{w}}^{(t)} - \eta_t \mathbf{g}_t$ and $\mathbb{E}\mathbf{g}_t = \overline{\mathbf{g}}_t$. Notably, for any $t \geq 0$, there exists a $t_0 \leq t$, such that $t - t_0 \leq \tau - 1$ and $\mathbf{w}_i^{(t_0)} = \overline{\mathbf{w}}^{(t_0)}$ for all $i = 1, 2, \ldots, M$. In this case, $\mathbf{x}_i^{(t)} = F_i(\mathbf{w}_i^{(t)}) = F(\mathbf{w}_i^{(t)} - \overline{\mathbf{w}}^{(t_0)}) + \overline{\mathbf{w}}^{(t_0)}$. Therefore, we have $\mathbb{E}\|\mathbf{x}_i^{(t)} - \mathbf{w}_i^{(t)}\|^2 \leq C_{\text{err}}^2 \|\mathbf{w}_i^{(t)} - \overline{\mathbf{w}}^{(t_0)}\|^2$.

*Proof of Theorem G.3.* Let $\Delta_t = \mathbb{E}\|\overline{\mathbf{w}}^{(t)} - \mathbf{w}^*\|^2$. From lemma H.6-H.8, it follows that

$$\Delta_{t+1} \leq (1 - \eta_t \mu)\Delta_t + \eta_t^2 B, \tag{55}$$

where

$$B = \sum_{i=1}^M p_i^2 \sigma_i^2 + 6L\Gamma + 16\left(1 + C_{\text{err}}^2\right)(\tau - 1)^2 G^2. \tag{56}$$

For a diminishing stepsize, $\eta_t = \frac{\beta}{t+\gamma}$ for some $\beta > \frac{1}{\mu}$ and $\gamma > 0$ such that $\eta_1 \leq \min\{\frac{1}{\mu}, \frac{1}{4L}\} = \frac{1}{4L}$ and $\eta_t \leq 2\eta_{t+\tau}$. We will prove $\Delta_t \leq \frac{v}{t+\gamma}$ where $v = \max\{\frac{\beta^2 B}{\beta\mu - 1}, (\gamma + 1)\Delta_1\}$. We prove it by induction. Firstly, the definition of $v$ ensures that it holds for $t = 1$. Assume the conclusion holds for some $t$, it follows that

$$\begin{aligned}
\Delta_{t+1} &\leq (1 - \eta_t \mu)\Delta_t + \eta_t^2 B \\
&\leq \left(1 - \frac{\beta\mu}{t+\gamma}\right)\frac{v}{t+\gamma} + \frac{\beta^2 B}{(t+\gamma)^2} \\
&= \frac{t+\gamma - 1}{(t+\gamma)^2}v + \frac{\beta^2 B}{(t+\gamma)^2} - \frac{\beta\mu - 1}{(t+\gamma)^2}v \\
&\leq \frac{v}{t+\gamma + 1}.
\end{aligned} \tag{57}$$

Then by the $L$-smoothness of $\mathcal{L}$,

$$\mathbb{E}\left[\mathcal{L}\left(\overline{\mathbf{w}}^{(t)}\right)\right] - \mathcal{L}^* \leq \frac{L}{2}\Delta_t \leq \frac{L}{2}\frac{v}{\gamma + t}. \tag{58}$$

Specifically, if we choose $\beta = \frac{2}{\mu}, \gamma = \max\{8\frac{L}{\mu}, \tau\} - 1$ and denote $\kappa = \frac{L}{\mu}$, then $\eta_t = \frac{2}{\mu}\frac{1}{\eta + t}$. One can verify that the choice of $\eta_t$ satisfies $\eta_t \leq 2\eta_{t+\tau}$ for $t \geq 1$. Then, we have

$$v = \max\left\{\frac{\beta^2 B}{\beta\mu - 1}, (\gamma + 1)\Delta_1\right\} \leq \frac{\beta^2 B}{\beta\mu - 1} + (\gamma + 1)\Delta_1 \leq \frac{4B}{\mu^2} + (\gamma + 1)\Delta_1, \tag{59}$$

and

$$\mathbb{E}\left[\mathcal{L}\left(\overline{\mathbf{w}}^{(t)}\right)\right] - \mathcal{L}^* \leq \frac{L}{2}\frac{v}{\gamma + t} \leq \frac{\kappa}{\gamma + t}\left(\frac{2B}{\mu} + \frac{\mu(\gamma + 1)}{2}\Delta_1\right). \tag{60}$$

$$\square$$

### H.2.1. LEMMAS

To convey our proof clearly, it would be necessary to prove certain useful lemmas. We defer the proof of these lemmas to latter section and focus on proving the main theorem.

**Lemma H.6.** *(Results of one step SGD). Assume assumption H.1 and H.2. If $\eta_t \leq \frac{1}{4L}$, we have*

$$\mathbb{E} \left\| \overline{\mathbf{v}}^{(t+1)} - \mathbf{w}^* \right\|^2 \leq (1 - \eta_t \mu) \, \mathbb{E} \left\| \overline{\mathbf{w}}^{(t)} - \mathbf{w}^* \right\|^2 + \eta_t^2 \mathbb{E} \left\| \mathbf{g}_t - \overline{\mathbf{g}}_t \right\|^2 + 6L\eta_t^2 \Gamma + 2\mathbb{E} \sum_{i=1}^{M} p_i \left\| \overline{\mathbf{w}}^{(t)} - \mathbf{x}_i^{(t)} \right\|^2 ,$$

*where $\Gamma = \mathcal{L}^* - \sum_{i=1}^{M} p_i \mathcal{L}_i^* \geq 0$.*

**Lemma H.7.** *(Bounding the variance). Assume assumption H.3 holds. It follows that*

$$\mathbb{E} \left\| \mathbf{g}_t - \overline{\mathbf{g}}_t \right\|^2 \leq \sum_{i=1}^{M} p_i^2 \sigma_i^2.$$

**Lemma H.8.** *(Bounding the divergence of $\{\mathbf{x}_i^{(t)}\}$). Assume assumption H.4 and H.5, that $\eta_t$ is non-increasing and $\eta_t \leq 2\eta_{t+\tau}$ for all $t \geq 0$. It follows that*

$$\mathbb{E} \left[ \sum_{i=1}^{M} p_i \left\| \overline{\mathbf{w}}^{(t)} - \mathbf{x}_i^{(t)} \right\|^2 \right] \leq 8 \left( 1 + C_{\text{err}}^2 \right) \eta_t^2 (\tau - 1)^2 G^2.$$

### H.2.2. THE PROOFS OF KEY LEMMAS

*Proof of Lemma H.6.* Notice that $\overline{\mathbf{v}}^{(t+1)} = \mathbf{w}^{(t)} - \eta_t \mathbf{g}_t$, then

$$\left\| \overline{\mathbf{v}}^{(t+1)} - \mathbf{w}^* \right\|^2 = \left\| \overline{\mathbf{w}}^{(t)} - \eta_t \mathbf{g}_t - \mathbf{w}^* - \eta_t \overline{\mathbf{g}}_t + \eta_t \overline{\mathbf{g}}_t \right\|^2$$
$$= \underbrace{\left\| \overline{\mathbf{w}}^{(t)} - \mathbf{w}^* - \eta_t \overline{\mathbf{g}}_t \right\|^2}_{A_1} + \underbrace{2\eta_t \left\langle \overline{\mathbf{w}}^{(t)} - \mathbf{w}^* - \eta_t \overline{\mathbf{g}}_t, \overline{\mathbf{g}}_t - \mathbf{g}_t \right\rangle}_{A_2} + \eta_t^2 \left\| \mathbf{g}_t - \overline{\mathbf{g}}_t \right\|^2. \tag{61}$$

Note that $\mathbb{E}A_2 = 0$. We next focus on bounding $A_1$. Again we split $A_1$ into three terms:

$$\left\| \overline{\mathbf{w}}^{(t)} - \mathbf{w}^* - \eta_t \overline{\mathbf{g}}_t \right\|^2 = \left\| \overline{\mathbf{w}}^{(t)} - \mathbf{w}^* \right\|^2 \underbrace{- 2\eta_t \left\langle \overline{\mathbf{w}}^{(t)} - \mathbf{w}^*, \overline{\mathbf{g}}_t \right\rangle}_{B_1} + \underbrace{\eta_t^2 \left\| \overline{\mathbf{g}}_t \right\|^2}_{B_2}. \tag{62}$$

From the the L-smoothness of $\mathcal{L}_i(\cdot)$, it follows that

$$\left\| \nabla \mathcal{L}_i \left( \mathbf{x}_i^{(t)} \right) \right\|^2 \leq 2L \left( \mathcal{L}_i \left( \mathbf{x}_i^{(t)} \right) - \mathcal{L}_i^* \right). \tag{63}$$

By the convexity of $\| \cdot \|^2$ and equ. (63), we have

$$B_2 = \eta_t^2 \left\| \overline{\mathbf{g}}_t \right\|^2 \leq \eta_t^2 \sum_{i=1}^{M} p_i \left\| \nabla \mathcal{L}_i \left( \mathbf{x}_i^{(t)} \right) \right\|^2 \leq 2L\eta_t^2 \sum_{i=1}^{M} p_i \left( \mathcal{L}_i \left( \mathbf{x}_i^{(t)} \right) - \mathcal{L}_i^* \right). \tag{64}$$

Note that

$$B_1 = -2\eta_t \left\langle \overline{\mathbf{w}}^{(t)} - \mathbf{w}^*, \overline{\mathbf{g}}_t \right\rangle = -2\eta_t \sum_{i=1}^{M} p_i \left\langle \overline{\mathbf{w}}^{(t)} - \mathbf{w}^*, \nabla \mathcal{L}_i \left( \mathbf{x}_i^{(t)} \right) \right\rangle$$
$$= -2\eta_t \sum_{i=1}^{M} p_i \left\langle \overline{\mathbf{w}}^{(t)} - \mathbf{x}_i^{(t)}, \nabla \mathcal{L}_i \left( \mathbf{x}_i^{(t)} \right) \right\rangle - 2\eta_t \sum_{i=1}^{M} p_i \left\langle \mathbf{x}_i^{(t)} - \mathbf{w}^*, \nabla \mathcal{L}_i \left( \mathbf{x}_i^{(t)} \right) \right\rangle. \tag{65}$$

By Cauchy-Schwarz inequality and AM-GM inequality, we have

$$-2\left\langle \overline{\mathbf{w}}^{(t)} - \mathbf{x}_i^{(t)}, \nabla \mathcal{L}_i\left(\mathbf{x}_i^{(t)}\right)\right\rangle \leq \frac{1}{\eta_t}\left\|\overline{\mathbf{w}}^{(t)} - \mathbf{x}_i^{(t)}\right\|^2 + \eta_t\left\|\nabla \mathcal{L}_i\left(\mathbf{x}_i^{(t)}\right)\right\|^2. \tag{66}$$

By the $\mu$-strong convexity of $\mathcal{L}_i(\cdot)$, we have

$$-\left\langle \mathbf{x}_i^{(t)} - \mathbf{w}^*, \nabla \mathcal{L}_i\left(\mathbf{x}_i^{(t)}\right)\right\rangle \leq -\left(\mathcal{L}_i\left(\mathbf{x}_i^{(t)}\right) - \mathcal{L}_i\left(\mathbf{w}^*\right)\right) - \frac{\mu}{2}\left\|\mathbf{x}_i^{(t)} - \mathbf{w}^*\right\|^2. \tag{67}$$

By combining equ. (62), equ. (65), equ. (66) and equ. (67), it follows that

$$
\begin{aligned}
A_1 = \left\|\overline{\mathbf{w}}^{(t)} - \mathbf{w}^* - \eta_t \overline{\mathbf{g}}_t\right\|^2 &\leq \left\|\overline{\mathbf{w}}^{(t)} - \mathbf{w}^*\right\|^2 + 2L\eta_t^2\sum_{i=1}^{M} p_i\left(\mathcal{L}_i\left(\mathbf{x}_i^{(t)}\right) - \mathcal{L}_i^*\right) \\
&\quad + \eta_t\sum_{i=1}^{M} p_i\left(\frac{1}{\eta_t}\left\|\overline{\mathbf{w}}^{(t)} - \mathbf{x}_i^{(t)}\right\|^2 + \eta_t\left\|\nabla \mathcal{L}_i\left(\mathbf{x}_i^{(t)}\right)\right\|^2\right) \\
&\quad - 2\eta_t\sum_{i=1}^{M} p_i\left(\mathcal{L}_i\left(\mathbf{x}_i^{(t)}\right) - \mathcal{L}_i\left(\mathbf{w}^*\right) + \frac{\mu}{2}\left\|\mathbf{x}_i^{(t)} - \mathbf{w}^*\right\|^2\right) \\
&\leq (1 - \mu\eta_t)\left\|\overline{\mathbf{w}}^{(t)} - \mathbf{w}^*\right\|^2 + \sum_{i=1}^{M} p_i\left\|\overline{\mathbf{w}}^{(t)} - \mathbf{x}_i^{(t)}\right\|^2 \\
&\quad + \underbrace{4L\eta_t^2\sum_{i=1}^{M} p_i\left(\mathcal{L}_i\left(\mathbf{x}_i^{(t)}\right) - \mathcal{L}_i^*\right) - 2\eta_t\sum_{i=1}^{M} p_i\left(\mathcal{L}_i\left(\mathbf{x}_i^{(t)}\right) - \mathcal{L}_i\left(\mathbf{w}^*\right)\right)}_{C},
\end{aligned}
\tag{68}
$$

where we use equ. (63) again and the inequality

$$
\begin{aligned}
-\mathbb{E}\left\|\mathbf{x}_i^{(t)} - \mathbf{w}^*\right\|^2 &= -\mathbb{E}\left\|\overline{\mathbf{w}}^{(t)} - \mathbf{x}_i^{(t)}\right\|^2 - \mathbb{E}\left\|\overline{\mathbf{w}}^{(t)} - \mathbf{w}^*\right\|^2 + 2\mathbb{E}\left\langle \overline{\mathbf{w}}^{(t)} - \mathbf{x}_i^{(t)}, \overline{\mathbf{w}}^{(t)} - \mathbf{w}^*\right\rangle \\
&\leq -\mathbb{E}\left\|\overline{\mathbf{w}}^{(t)} - \mathbf{w}^*\right\|^2 + 2\mathbb{E}\left\langle \overline{\mathbf{w}}^{(t)} - \mathbf{w}_i^{(t)} + \mathbf{w}_i^{(t)} - \mathbf{x}_i^{(t)}, \overline{\mathbf{w}}^{(t)} - \mathbf{w}^*\right\rangle \\
&= -\mathbb{E}\left\|\overline{\mathbf{w}}^{(t)} - \mathbf{w}^*\right\|^2,
\end{aligned}
\tag{69}
$$

where $\mathbb{E}\mathbf{x}_i^{(t)} = \mathbb{E}\mathbf{w}_i^{(t)}$ is based on $\mathbf{x}_i^{(t)} = F_i(\mathbf{w}_i^{(t)}) = F(\mathbf{w}_i^{(t)} - \overline{\mathbf{w}}^{(t_0)}) + \overline{\mathbf{w}}^{(t_0)}$. We next aim to bound C. We define $\gamma_t = 2\eta_t(1 - 2L\eta_t)$. Since $\eta_t \leq \frac{1}{4L}, \eta_t \leq \gamma_t \leq 2\eta_t$. Then we split C into two terms:

$$
\begin{aligned}
C &= -2\eta_t(1 - 2L\eta_t)\sum_{i=1}^{M} p_i\left(\mathcal{L}_i\left(\mathbf{x}_i^{(t)}\right) - \mathcal{L}_i^*\right) + 2\eta_t\sum_{i=1}^{M} p_i\left(\mathcal{L}_i\left(\mathbf{w}^*\right) - \mathcal{L}_i^*\right) \\
&= -\gamma_t\sum_{i=1}^{M} p_i\left(\mathcal{L}_i\left(\mathbf{x}_i^{(t)}\right) - \mathcal{L}^*\right) + (2\eta_t - \gamma_t)\sum_{i=1}^{M} p_i\left(\mathcal{L}^* - \mathcal{L}_i^*\right) \\
&= \underbrace{-\gamma_t\sum_{i=1}^{M} p_i\left(\mathcal{L}_i\left(\mathbf{x}_i^{(t)}\right) - \mathcal{L}^*\right)}_{D} + 4L\eta_t^2\Gamma,
\end{aligned}
\tag{70}
$$

where in the last equation, we use the notation $\Gamma = \sum_{i=1}^{M} p_i \left( \mathcal{L}^* - \mathcal{L}_i^* \right) = \mathcal{L}^* - \sum_{i=1}^{M} p_i \mathcal{L}_i^*$. To bound $D$, we have

$$
\begin{aligned}
\sum_{i=1}^{M} p_i \left( \mathcal{L}_i \left( \mathbf{x}_i^{(t)} \right) - \mathcal{L}^* \right) &= \sum_{i=1}^{M} p_i \left( \mathcal{L}_i \left( \mathbf{x}_i^{(t)} \right) - \mathcal{L}_i \left( \overline{\mathbf{w}}^{(t)} \right) \right) + \sum_{i=1}^{M} p_i \left( \mathcal{L}_i \left( \overline{\mathbf{w}}^{(t)} \right) - \mathcal{L}^* \right) \\
&\geq \sum_{i=1}^{M} p_i \left\langle \nabla \mathcal{L}_i \left( \overline{\mathbf{w}}^{(t)} \right), \mathbf{x}_i^{(t)} - \overline{\mathbf{w}}^{(t)} \right\rangle + \left( \mathcal{L} \left( \overline{\mathbf{w}}^{(t)} \right) - \mathcal{L}^* \right) \\
&\geq -\frac{1}{2} \sum_{i=1}^{M} p_i \left[ \eta_t \left\| \nabla \mathcal{L}_i \left( \overline{\mathbf{w}}^{(t)} \right) \right\|^2 + \frac{1}{\eta_t} \left\| \mathbf{x}_i^{(t)} - \overline{\mathbf{w}}^{(t)} \right\|^2 \right] + \left( \mathcal{L} \left( \overline{\mathbf{w}}^{(t)} \right) - \mathcal{L}^* \right) \\
&\geq -\sum_{i=1}^{M} p_i \left[ \eta_t L \left( \mathcal{L}_i \left( \overline{\mathbf{w}}^{(t)} \right) - \mathcal{L}_i^* \right) + \frac{1}{2\eta_t} \left\| \mathbf{x}_i^{(t)} - \overline{\mathbf{w}}^{(t)} \right\|^2 \right] + \left( \mathcal{L} \left( \overline{\mathbf{w}}^{(t)} \right) - \mathcal{L}^* \right),
\end{aligned}
\tag{71}
$$

where the first inequality results from the convexity of $\mathcal{L}_i(\cdot)$, the second inequality from AM-GM inequality and the third inequality from equ. (63). Therefore

$$
\begin{aligned}
C &= \gamma_t \sum_{i=1}^{M} p_i \left[ \eta_t L \left( \mathcal{L}_i \left( \overline{\mathbf{w}}^{(t)} \right) - \mathcal{L}_i^* \right) + \frac{1}{2\eta_t} \left\| \mathbf{x}_i^{(t)} - \overline{\mathbf{w}}^{(t)} \right\|^2 \right] - \gamma_t \left( \mathcal{L} \left( \overline{\mathbf{w}}^{(t)} \right) - \mathcal{L}^* \right) + 4L\eta_t^2 \Gamma \\
&= \gamma_t \left( \eta_t L - 1 \right) \sum_{i=1}^{M} p_i \left( \mathcal{L}_i \left( \overline{\mathbf{w}}^{(t)} \right) - \mathcal{L}^* \right) + \left( 4L\eta_t^2 + \gamma_t \eta_t L \right) \Gamma + \frac{\gamma_t}{2\eta_t} \sum_{i=1}^{M} p_i \left\| \mathbf{x}_i^{(t)} - \overline{\mathbf{w}}^{(t)} \right\|^2 \\
&\leq 6L\eta_t^2 \Gamma + \sum_{i=1}^{M} p_i \left\| \mathbf{x}_i^{(t)} - \overline{\mathbf{w}}^{(t)} \right\|^2,
\end{aligned}
\tag{72}
$$

where in the last inequality, we use the following facts: (1) $\eta_t L - 1 \leq -\frac{3}{4} \leq 0$ and $\sum_{i=1}^{M} p_i (\mathcal{L}_i(\overline{\mathbf{w}}^{(t)}) - \mathcal{L}^*) = \mathcal{L}(\overline{\mathbf{w}}^{(t)}) - \mathcal{L}^* \geq 0$ (2) $\Gamma \geq 0$ and $4L\eta_t^2 + \gamma_t \eta_t L \leq 6\eta_t^2 L$ and (3) $\frac{\gamma_t}{2\eta_t} \leq 1$. Recalling the expression of $A_1$ and plugging $C$ into it, we have

$$
\begin{aligned}
A_1 &= \left\| \overline{\mathbf{w}}^{(t)} - \mathbf{w}^* - \eta_t \overline{\mathbf{g}}_t \right\|^2 \\
&\leq \left( 1 - \mu\eta_t \right) \left\| \overline{\mathbf{w}}^{(t)} - \mathbf{w}^* \right\|^2 + 2 \sum_{i=1}^{M} p_i \left\| \overline{\mathbf{w}}^{(t)} - \mathbf{x}_i^{(t)} \right\|^2 + 6\eta_t^2 L\Gamma.
\end{aligned}
\tag{73}
$$

Using equ. (73) and taking expectation on both sides of equ. (61), we erase the randomness from stochastic gradients, we complete the proof. □

*Proof of Lemma H.7.* From assumption H.3, the variance of the stochastic gradients in device $i$ is bounded by $\sigma_i^2$, then

$$
\begin{aligned}
\mathbb{E} \left\| \mathbf{g}_t - \overline{\mathbf{g}}_t \right\|^2 &= \mathbb{E} \left\| \sum_{i=1}^{M} p_i \left( \nabla \mathcal{L}_i \left( \mathbf{w}_i^{(t)}, \zeta_i^{(t)} \right) - \nabla \mathcal{L}_i \left( \mathbf{w}_i^{(t)} \right) \right) \right\|^2 \\
&= \sum_{i=1}^{M} p_i^2 \mathbb{E} \left\| \nabla \mathcal{L}_i \left( \mathbf{w}_i^{(t)}, \zeta_i^{(t)} \right) - \nabla \mathcal{L}_i \left( \mathbf{w}_i^{(t)} \right) \right\|^2 \\
&\leq \sum_{i=1}^{M} p_i^2 \sigma_i^2.
\end{aligned}
\tag{74}
$$

□

*Proof of Lemma H.8.* We have:

$$\sum_{i=1}^{M} p_i \mathbb{E} \left\| \overline{\mathbf{w}}^{(t)} - \mathbf{x}_i^{(t)} \right\|^2 = \sum_{i=1}^{M} p_i \mathbb{E} \left\| \left( \overline{\mathbf{w}}^{(t)} - \mathbf{w}_i^{(t)} \right) - \left( \mathbf{w}_i^{(t)} - \mathbf{x}_i^{(t)} \right) \right\|^2$$

$$\leq 2 \sum_{i=1}^{M} p_i \mathbb{E} \left\| \overline{\mathbf{w}}^{(t)} - \mathbf{w}_i^{(t)} \right\|^2 + 2 \sum_{i=1}^{M} p_i \mathbb{E} \left\| \mathbf{w}_i^{(t)} - \mathbf{x}_i^{(t)} \right\|^2. \tag{75}$$

Since FedAvg requires a communication each $\tau$ steps. Therefore, for any $t \geq 0$, there exists a $t_0 \leq t$, such that $t - t_0 \leq \tau - 1$ and $\mathbf{w}_i^{(t_0)} = \mathbf{w}^{(t_0)}$ for all $k = 1, 2, \ldots, M$. Also, we use the fact that $\eta_t$ is non-increasing and $\eta_{t_0} \leq 2\eta_t$ for all $t - t_0 \leq \tau - 1$, then

$$\sum_{i=1}^{M} p_i \mathbb{E} \left\| \overline{\mathbf{w}}^{(t)} - \mathbf{w}_i^{(t)} \right\|^2 = \sum_{i=1}^{M} p_i \mathbb{E} \left\| \left( \mathbf{w}_i^{(t)} - \overline{\mathbf{w}}^{(t_0)} \right) - \left( \overline{\mathbf{w}}^{(t)} - \overline{\mathbf{w}}^{(t_0)} \right) \right\|^2$$

$$\leq \sum_{i=1}^{M} p_i \mathbb{E} \left\| \mathbf{w}_i^{(t)} - \overline{\mathbf{w}}^{(t_0)} \right\|^2$$

$$\leq \sum_{i=1}^{M} p_i \mathbb{E} \sum_{t_0}^{t-1} (\tau - 1)\eta_t^2 \left\| \nabla \mathcal{L}_i \left( \mathbf{w}_i^{(t)}, \zeta_i^{(t)} \right) \right\|^2 \tag{76}$$

$$\leq \sum_{i=1}^{M} p_i \sum_{t_0}^{t-1} (\tau - 1)(\eta_{t_0})^2 G^2$$

$$\leq (\eta_{t_0})^2 (\tau - 1)^2 G^2$$

$$\leq 4\eta_t^2 (\tau - 1)^2 G^2.$$

Here in the first inequality, we use $\mathbb{E}\|X - \mathbb{E}X\|^2 \leq \mathbb{E}\|X\|^2$ where $X = \mathbf{w}_i^{(t)} - \mathbf{w}^{(t_0)}$ with probability $p_i$. In the second inequality, we use Jensen inequality:

$$\left\| \mathbf{w}_i^{(t)} - \overline{\mathbf{w}}^{(t_0)} \right\|^2 = \left\| \sum_{t_0}^{t-1} \eta_t \nabla \mathcal{L}_i \left( \mathbf{w}_i^{(t)}, \zeta_i^{(t)} \right) \right\|^2 \leq (t - t_0) \sum_{t_0}^{t-1} \eta_t^2 \left\| \nabla \mathcal{L}_i \left( \mathbf{w}_i^{(t)}, \zeta_i^{(t)} \right) \right\|^2. \tag{77}$$

In the third inequality, we use $\eta_t \leq \eta_{t_0}$ for $t \geq t_0$ and $\mathbb{E}\|\nabla \mathcal{L}_i(w_i^{(t)}, \zeta_i^{(t)})\|^2 \leq G^2$ for $k = 1, 2, \ldots, M$ and $t \geq 1$. In the last inequality, we use $\eta_{t_0} \leq 2\eta_{t_0+\tau} \leq 2\eta_t$ for $t_0 \leq t \leq t_0 + \tau$.

According to assumption H.5, we have $\mathbb{E} \left\| \mathbf{w}_i^{(t)} - \mathbf{x}_i^{(t)} \right\|^2 \leq C_{\text{err}}^2 \mathbb{E} \left\| \mathbf{w}_i^{(t)} - \overline{\mathbf{w}}^{(t_0)} \right\|^2$ as discussed in Section H.2. Then the second term in Equ. (75) can be bounded by reusing the result in Equ. (76) as

$$\sum_{i=1}^{M} p_i \mathbb{E} \left\| \mathbf{w}_i^{(t)} - \mathbf{x}_i^{(t)} \right\|^2 \leq C_{\text{err}}^2 \sum_{i=1}^{M} p_i \mathbb{E} \left\| \mathbf{w}_i^{(t)} - \overline{\mathbf{w}}^{(t_0)} \right\|^2$$

$$\leq 4 C_{\text{err}}^2 \eta_t^2 (\tau - 1)^2 G^2. \tag{78}$$

Plugging Equ. (76) and Equ. (78) into Equ. (75), we have the result in lemma H.8

$$\sum_{i=1}^{M} p_i \mathbb{E} \left\| \overline{\mathbf{w}}^{(t)} - \mathbf{x}_i^{(t)} \right\|^2 \leq 8 \left( 1 + C_{\text{err}}^2 \right) \eta_t^2 (\tau - 1)^2 G^2. \tag{79}$$

$\square$

## H.3. The Proof of Theorem G.4

We analyze Fed-Kalter in the setting of partial device participation.

Recall that $\mathbf{w}_i^{(t)}$ is the model parameter maintained in the i-th device at the t-th step. $\mathcal{I}_\tau = \{n\tau | n = 1, 2, \ldots, M\}$ is the set of global synchronization steps. Again, $\overline{\mathbf{g}}_t = \sum_{i=1}^M p_i \nabla \mathcal{L}_i(\mathbf{x}_i^{(t)})$ and $\mathbf{g}_t = \sum_{i=1}^M p_i \nabla \mathcal{L}_i(\mathbf{x}_i^{(t)}, \zeta_i^{(t)})$. Therefore, $\overline{\mathbf{v}}^{(t+1)} = \overline{\mathbf{w}}^{(t)} - \gamma_t \mathbf{g}_t$ and $\mathbb{E}\mathbf{g}_t = \overline{\mathbf{g}}_t$.

Now we consider the case where Fed-Kalter samples a random set $\mathbb{S}_t$ of devices to participate in each round of training. This make the analysis a little bit intricate, since $\mathbb{S}_t$ varies each $\tau$ steps. Following ((Li et al., 2020b; 2024a)), we assume that Fed-Kalter always activates all devices at the beginning of each round and then uses the parameters maintained in only a few sampled devices to produce the next-round parameter. It is clear that this updating scheme is equivalent to the original. As assumed in theorem G.4 that $p_1 = \cdots = p_M = \frac{1}{M}$, the update of Fed-Kalter with partial devices active can be described as: for all $i \in [M]$,

$$
\begin{aligned}
\mathbf{v}_i^{(t+1)} &= \mathbf{w}_i^{(t)} - \eta_t \nabla \mathcal{L}_i \left( \mathbf{x}_i^{(t)}, \zeta_i^{(t)} \right) \\
\mathbf{x}_i^{(t)} &= F_i \left( \mathbf{w}_i^{(t)} \right) \\
\mathbf{w}_i^{(t+1)} &= \begin{cases} \mathbf{v}_i^{(t+1)} & \text{if } t + 1 \notin \mathcal{I}_\tau \\ \frac{\sum_{i\in\mathbb{S}_{t+1}} p_i \mathbf{v}_i^{(t)}}{\sum_{i\in\mathbb{S}_{t+1}} p_i} = \frac{1}{N} \sum_{i\in\mathbb{S}_{t+1}} \mathbf{v}_i^{(t+1)} & \text{if } t + 1 \in \mathcal{I}_\tau \end{cases}.
\end{aligned}
\tag{80}
$$

### H.3.1. LEMMAS

**Lemma H.9.** *(Bounding the variance of $\overline{\mathbf{w}}^{(t)}$). In the case of partial device participation in theorem G.4, with assumption H.4, assume that $\gamma_t$ is non-increasing and $\gamma_t \leq \gamma_{t+\tau}$ for all $t \geq 0$. It follows that*

$$
\mathbb{E} \left\| \overline{\mathbf{w}}^{(t+1)} - \overline{\mathbf{v}}^{(t+1)} \right\|^2 \leq 4 \frac{M - N}{N(M - 1)} \eta_t^2 \tau^2 G^2.
\tag{81}
$$

### H.3.2. COMPLETE PROOF OF THEOREM G.4

Let $\Delta_t = \mathbb{E}\|\overline{\mathbf{w}}^{(t)} - \mathbf{w}^*\|^2$. From lemma H.6, H.7, H.8 and H.9, it follows that

$$
\Delta_{t+1} \leq (1 - \eta_t \mu) \Delta_t + \eta_t^2 B,
\tag{82}
$$

where

$$
B = \frac{1}{M} \sum_{i=1}^M \sigma_i^2 + 6L\Gamma + 16 \left( 1 + C_{\text{err}}^2 \right) (\tau - 1)^2 G^2 + 4 \frac{M - N}{N(M - 1)} \tau^2 G^2.
\tag{83}
$$

The only difference between Equ.(82) and Equ.(55) is the value of constant B. Following the same process, we can get the result of theorem G.4

$$
\mathbb{E} \left[ \mathcal{L} \left( \overline{\mathbf{w}}^{(t)} \right) \right] - \mathcal{L}^* \leq \frac{L}{2} \frac{v}{\gamma + t} \leq \frac{\kappa}{\gamma + t} \left( \frac{2B}{\mu} + \frac{\mu(\gamma + 1)}{2} \left\| \mathbf{w}^{(1)} - \mathbf{w}^* \right\|^2 \right),
\tag{84}
$$

where $\gamma = \max\{8\frac{L}{\mu}, \tau\} - 1$ and $\kappa = \frac{L}{\mu}$.

### H.3.3. THE PROOFS OF KEY LEMMAS

*Proof of Lemma H.9.* Notice that $\overline{\mathbf{w}}^{(t+1)} = \overline{\mathbf{v}}^{(t+1)}$ when $t + 1 \notin \mathcal{I}_\tau$ and $\overline{\mathbf{w}}^{(t+1)} = \frac{1}{N} \sum_{i\in\mathbb{S}_{t+1}} \mathbf{v}_i^{(t+1)}$, $\overline{\mathbf{v}}^{(t+1)} = \sum_{i=1}^M p_i \mathbf{v}^{(t+1)}$ when $t + 1 \in \mathcal{I}_\tau$. Hence, we have

$$
\mathbb{E} \left\| \overline{\mathbf{w}}^{(t+1)} - \overline{\mathbf{v}}^{(t+1)} \right\|^2 = \mathbb{E} \underbrace{\left\| \frac{1}{N} \sum_{i\in\mathbb{S}_{t+1}} \mathbf{v}_i^{(t+1)} - \overline{\mathbf{v}}^{(t+1)} \right\|^2}_{A}.
\tag{85}
$$

To bound A, we have

$$
\begin{aligned}
A &= \mathbb{E} \left\| \frac{1}{N} \sum_{i \in \mathbb{S}_{t+1}} \mathbf{v}_i^{(t+1)} - \overline{\mathbf{v}}^{(t+1)} \right\|^2 \\
&= \frac{1}{N^2} \mathbb{E} \left\| \sum_{i=1}^{M} \mathbb{I} \{ i \in \mathbb{S}_{t+1} \} \left( \mathbf{v}_i^{(t+1)} - \overline{\mathbf{v}}^{(t+1)} \right) \right\|^2 \\
&= \frac{1}{N^2} \mathbb{E} \left[ \sum_{i=1}^{M} \mathbb{P} \left( i \in \mathbb{S}_{t+1} \right) \left\| \mathbf{v}_i^{(t+1)} - \overline{\mathbf{v}}^{(t+1)} \right\|^2 + \sum_{i \neq j} \mathbb{P} \left( i, j \in \mathbb{S}_{t+1} \right) \left\langle \mathbf{v}_i^{(t+1)} - \overline{\mathbf{v}}^{(t+1)}, \mathbf{v}_j^{(t+1)} - \overline{\mathbf{v}}^{(t+1)} \right\rangle \right] \quad (86) \\
&= \frac{1}{NM} \mathbb{E} \sum_{i=1}^{M} \left\| \mathbf{v}_i^{(t+1)} - \overline{\mathbf{v}}^{(t+1)} \right\|^2 + \frac{N-1}{NM(M-1)} \mathbb{E} \sum_{i \neq j} \left\langle \mathbf{v}_i^{(t+1)} - \overline{\mathbf{v}}^{(t+1)}, \mathbf{v}_j^{(t+1)} - \overline{\mathbf{v}}^{(t+1)} \right\rangle \\
&= \frac{M-N}{NM(M-1)} \sum_{i=1}^{M} \mathbb{E} \left\| \mathbf{v}_i^{(t+1)} - \overline{\mathbf{v}}^{(t+1)} \right\|^2,
\end{aligned}
$$

where we use the following equalities: (1) $\mathbb{P}(i \in \mathbb{S}_{t+1}) = \frac{N}{M}$ and $\mathbb{P}(i, j \in \mathbb{S}_{t+1}) = \frac{N(N-1)}{M(M-1)}$ for all $i \neq j$ and (2)$\sum_{i=1}^{M} \| \mathbf{v}_i^{(t+1)} - \overline{\mathbf{v}}^{(t+1)} \|^2 + \sum_{i \neq j} \left\langle \mathbf{v}_i^{(t+1)} - \overline{\mathbf{v}}^{(t+1)}, \mathbf{v}_j^{(t+1)} - \overline{\mathbf{v}}^{(t+1)} \right\rangle = 0$. We have

$$
\begin{aligned}
A &= \frac{M-N}{NM(M-1)} \sum_{i=1}^{M} \mathbb{E} \left\| \mathbf{v}_i^{(t+1)} - \overline{\mathbf{v}}^{(t+1)} \right\|^2 \\
&\leq \frac{M-N}{N(M-1)} 4\eta_t^2 \tau^2 G^2.
\end{aligned}
\quad (87)
$$

Therefore, we have the result in lemma H.9

$$
\mathbb{E} \left\| \overline{\mathbf{w}}^{(t+1)} - \overline{\mathbf{v}}^{(t+1)} \right\|^2 \leq 4 \frac{M-N}{N(M-1)} \eta_t^2 \tau^2 G^2. \quad (88)
$$

$\square$

