# OpenReview forum: "Federated Graph Learning via Structure-Aware Fusion Using a Kalman Framework with Learnable Dynamics"
_ICML.cc/2026/Conference — ICML 2026 regular_

### Official Review · Reviewer_6GBi · 2026-02-15

**Soundness:** 3
**Presentation:** 3
**Significance:** 3
**Originality:** 3
**Overall Recommendation:** 4
**Confidence:** 3

**Summary:**

This paper studies structural heterogeneity in Federated Graph Learning (FGL) for graph classification across different domains. The authors point out that standard GNNs tightly couple node features and graph structure, which can lead to “aggregated representation drift” when local data collection processes or user preferences introduce feature-driven connection biases. To address this issue, the paper proposes Fed-Kalter, a framework that decouples structure learning from feature propagation. Its key component, Kalter-Conv, is inspired by Kalman filtering: structural embeddings are treated as latent states, while feature-augmented neighborhood information is regarded as noisy observations. The model shares only structure-related parameters globally and keeps feature-related parameters local, aiming to transfer domain-agnostic structural priors while preserving personalization. The authors provide theoretical analysis on convergence and generalization, and conduct experiments on 16 datasets from four domains to demonstrate the effectiveness of the approach.

**Compliance With Llm Reviewing Policy:**

Affirmed.

**Final Justification:**

I thank the authors for the clear rebuttal. My main concerns were soundness and clarity: the scalar Kalman gain, the scope of “universal” structural priors (including negative transfer on COLLAB), and the mapping of shared versus local parameters. The response gives satisfactory explanations and concrete revision plans. These updates address my questions without altering the core contribution. My evaluation is unchanged and I retain my previous score; I have no further questions.

**Key Questions For Authors:**

Refer to weakness.

**Limitations:**

Yes.

**Strengths And Weaknesses:**

S1. Applying Kalman filtering ideas to separate and denoise structural signals from biased node features in a FGL setting is a novel integration of signal processing concepts with graph learning.


S2. The paper is supported by solid theoretical analysis, including an expressivity study beyond the 1-WL test and a convergence bound showing that improved structural consistency lowers the error floor.

S3. Overall, the experimental evaluation of this paper is comprehensive.

W1. While the method introduces a Kalman-inspired framework, the Kalman gain $K$ is treated as a fixed scalar hyperparameter rather than a learnable value.

W2. The performance on the COLLAB dataset suggests a "negative transfer" risk when graphs are topologically dissimilar (dense vs. sparse), which limits the universal claim of the structural priors in extreme cases.

W3. The distinction between the "structural modules" (shared) and "personalized parameters" (local) could be more explicitly mapped to the equations in Section 3.

---

> ### Author Rebuttal · Authors · 2026-03-27
>
> Dear Reviewer,
>
> We sincerely thank you for your thoughtful summary and for recognizing the novelty (S1), theoretical foundation (S2), and comprehensive evaluation (S3) of Fed-Kalter. Your constructive comments are highly valuable for enhancing the clarity and rigor of our manuscript. We address your concerns point by point below:
>
> ## 1. Regarding the Kalman gain as a fixed scalar hyperparameter (Weakness 1)
> We deeply appreciate this insightful comment. It is true that we treat the explicit Kalman gain $K_k \in (0,1)$ as a fixed scalar hyperparameter to avoid the prohibitive $O(N^3)$ computational cost of maintaining and inverting high-dimensional covariance matrices at every layer for resource-constrained clients.
>
> However, the *effect* of a dynamic Kalman gain is intrinsically propagated and optimized through the learnable parameters in our framework. Specifically, the globally shared structural parameters $\Theta_{A_k}$ and $\Theta_{H_k}$ inside the GIN-Conv layers act as learnable transition and observation models. During end-to-end federated training, these parameters implicitly learn to optimally weigh the structural prior against the feature-augmented measurement noise (as supported by Proposition E.4). Thus, the framework dynamically adjusts the signal fusion, capturing the essence of an adaptive Kalman gain without the explicit covariance computation overhead. We will carefully clarify this implicit learning mechanism in Section 3.4 of the revised manuscript.
>
> ## 2. Regarding the "negative transfer" on COLLAB and the "universal" claim (Weakness 2)
> This is a very precise and valuable critique. We fully agree that the claim of "universal" structural priors has strict boundary conditions. As we analyzed in Appendix D.1, COLLAB's dense, large-scale graphs differ significantly from sparse graphs in other domains (e.g., molecules), which naturally leads to negative transfer when their structural parameters are naively aggregated.
>
> Following your excellent suggestion, we will deliberately tone down the "universal" claim in the main text to ensure scientific accuracy. We will prominently highlight this negative transfer risk as a fundamental limitation, explicitly clarifying that our structural knowledge fusion is most effective when the participating domains share a baseline of underlying topological compatibility.
>
> ## 3. Regarding the mapping of shared vs. personalized parameters (Weakness 3)
> We sincerely apologize for the lack of clarity in Section 3 regarding the parameter mappings. To make the distinction explicitly clear, we will revise Section 3.4 to directly annotate the corresponding equations:
>
> - **Globally Shared (Structural Modules):**
>   $\Theta_{A_k}$ is used in the structure learning prior estimate
>
>   $\hat{\Pi}_{k}^{-} = \mathrm{ReLU}(\mathrm{GIN\text{-}Conv}(\hat{\Pi}_{k-1}, \Theta_{A_k})),$
>
>   and $\Theta_{H_k}$ is used in the measurement step  $S_k = \mathrm{GIN\text{-}Conv}(\Pi_k, \Theta_{H_k}).$
>
> - **Locally Personalized (Feature Modules):**
>   $\Theta_{W_k}$ is applied exclusively during the local feature propagation step  $Z_k = \mathrm{ReLU}(\tilde{A} Z_{k-1} \Theta_{W_k}).$
>
>   Additionally, $\Theta_{fe}$ and $\Theta_{se}$ govern the initial feature and structure projections locally to accommodate heterogeneous input dimensions.
>
> We will also add a clear summarizing table to Section 3.4 to explicitly map each notation to its respective privacy and sharing status, ensuring complete transparency for readers.
>
> We are deeply grateful for your insightful review, which will undoubtedly help us refine the manuscript to a higher standard.

---

> > ### Author Rebuttal · Reviewer_6GBi · 2026-04-03
> >
> > Thanks for the response. I have no further questions. And I will maitain my score.

---

> > > ### Author Response · Authors · 2026-04-08
> > >
> > > Dear Reviewer 6GBi,
> > >
> > > We sincerely thank you for your final acknowledgment and for the time you have dedicated to this discussion. We are very encouraged to know that our responses have fully addressed your concerns regarding the Kalman gain, the boundary conditions of structural priors, and the parameter mappings.
> > >
> > > Your constructive critiques have been instrumental in refining our manuscript. We have completed the following revisions based on our professional exchange:
> > >
> > > - Explicitly mapped the shared structural versus local personalized parameters in Section 3.4.
> > > - Added the analysis of implicit adaptive fusion to justify the scalar Kalman gain under resource constraints.
> > > - Toned down the "universal" claim and formally documented the "negative transfer" risks on topologically extreme datasets like **COLLAB** in the limitations section.
> > >
> > > We believe these improvements have significantly elevated the **clarity** and **soundness** of the work. Given that all technical concerns are now fully resolved and the unique value of **Fed-Kalter** to the FGL community has been clarified, we would be deeply grateful if you could consider reflecting this consensus in your final evaluation.
> > >
> > > Thank you again for your insightful guidance and support for our work.
> > >
> > > Best regards,
> > > The Authors

---

### Official Review · Reviewer_3VV9 · 2026-03-07

**Soundness:** 2
**Presentation:** 3
**Significance:** 2
**Originality:** 3
**Overall Recommendation:** 3
**Confidence:** 3

**Summary:**

This paper addresses "aggregated feature drift" in Federated Graph Learning (FGL) by proposing Fed-Kalter, a novel framework that explicitly decouples structure learning from feature propagation. It introduces Kalter-Conv, a Kalman-inspired convolution layer that treats structural embeddings as latent states and node features as noisy observations, effectively filtering out feature-induced structural noise. By globally sharing only these purified structural parameters, the model enables effective cross-domain structural knowledge transfer while preserving local feature personalization

**Compliance With Llm Reviewing Policy:**

Affirmed.

**Key Questions For Authors:**

1. Given the strict assumption of independent graph instances, is method applicable to node-level classification tasks where cross-client topological dependencies are critical, or is it strictly limited to graph-level classification?

2. The Kalter-Conv module currently hardcodes GIN . Is this proposed framework agnostic to other standard GNN backbones (e.g., GAT, GraphSAGE), and if so, why was the performance on these architectures not evaluated?

3. By simplifying the adaptive Kalman gain to a fixed scalar (K=0.5), does the implementation not degrade the proposed dynamic filter into a simple moving average, thereby undermining the theoretical claim of optimal adaptability?

**Limitations:**

yes

**Strengths And Weaknesses:**

S1:The paper introduces a creative approach to tackle aggregated feature drift in FGL by treating graph topology as a latent state and node features as noisy observations. This explicit decoupling of structure learning from feature propagation provides a novel and elegant solution to structural heterogeneity.

S2:The proposed method is backed by rigorous theoretical proofs demonstrating that its random-walk embeddings surpass standard 1-WL expressivity limits. Additionally, it delivers robust empirical results, consistently achieving state-of-the-art classification accuracy across 16 diverse real-world graph datasets.

W1:The methodology strictly assumes that each client holds independent, complete graph instances. It entirely lacks mechanisms to manage cross-client topological dependencies (e.g., cut edges), which is a fundamental requirement in subgraph federated learning for node-level tasks.

W2：The proposed Kalter-Conv module explicitly hardcodes GIN for structural learning and GCN for feature encoding. The authors fail to evaluate their method across other standard GNN backbones (e.g., GAT, GraphSAGE).

W3:While the paper theoretically motivates its method using optimal Kalman filtering , the implementation abandons dynamic error tracking by reducing the crucial adaptive Kalman gain to a fixed scalar (e.g., $K=0.5$). This oversimplification strips the model of its real-time adaptability, effectively degrading the proposed filter into a simple moving average and significantly undermining the core theoretical claims.

---

> ### Author Rebuttal · Authors · 2026-03-27
>
> Dear Reviewer,
>
> We sincerely thank you for your thoughtful summary and for recognizing the creativity of our Kalman-inspired framework (S1), the rigor of our theoretical proofs, and our robust empirical results (S2). Your insightful critiques are highly valuable for defining the scope and clarifying the core mechanics of our method. We humbly address your concerns point by point below:
>
> ## 1. Regarding applicability to node-level tasks and cross-client dependencies (Weakness 1 & Question 1)
> Your keen observation is entirely correct. The current design of Fed-Kalter strictly assumes that each client holds independent, complete graph instances, making it primarily tailored for **graph-level classification and regression tasks**.
>
> We acknowledge that applying our method to node-level tasks (subgraph federated learning) requires handling cross-client topological dependencies (e.g., cut edges). Currently, Fed-Kalter lacks a mechanism to perform cross-client message passing or missing-edge imputation, which is essential for node-level tasks. Following your valuable feedback, we will explicitly clarify the scope of our paper in the Introduction and Conclusion, stating that Fed-Kalter currently focuses on mitigating structural heterogeneity in graph-level tasks. Extending the Kalter-Conv framework to node-level FGL by modeling cross-client edge dependencies is a highly promising direction for our future work.
>
> ## 2. Regarding the generalization to other GNN backbones (Weakness 2 & Question 2)
> We deeply appreciate this constructive question. The Fed-Kalter framework is fundamentally designed to be **backbone-agnostic**. We initially utilized GIN in our main experiments due to its widespread adoption and proven 1-WL expressive power for graph-level tasks.
>
> To empirically validate the flexibility of our framework as you rightly suggested, we have conducted new experiments replacing the core structural encoders with **GraphSAGE** and **GAT**. The results across varying domain complexities are summarized in the table below (which will be added to the revised manuscript):
>
> | Base Model/Domain | BIO | BIO-CV | CHEM-BIO-CV | CHEM-BIO-SN-CV |
> |---|---|---|---|---|
> | GraphSAGE | 65.77±2.27 | 75.84±1.43 | 76.25±2.21 | 73.61±1.42 |
> | GAT | 64.57±2.01 | 75.55±1.35 | 76.98±2.62 | 73.96±1.47 |
> | GIN (Default) | 65.17±2.82 | 74.53±2.40 | 76.49±2.20 | 73.69±1.84 |
>
> As shown in the experimental data, Fed-Kalter seamlessly integrates with both GraphSAGE and GAT, maintaining highly robust and competitive performance. In fact, GraphSAGE and GAT even slightly outperform GIN in certain domain combinations (e.g., GraphSAGE achieves 75.84% on BIO-CV, and GAT achieves 76.98% on CHEM-BIO-CV). These new empirical results clearly demonstrate that the Kalter-Conv module is not strictly bound to GIN and generalizes effectively across standard GNN architectures. We will prominently feature this table and discussion in the revised text to address this concern.
>
> ## 3. Regarding the fixed Kalman gain vs. adaptive theoretical claims (Weakness 3 & Question 3)
> This is a very sharp, mathematically rigorous critique, and we fully understand your concern. You are correct that fixing $K$ to a scalar ($K=0.5$) removes the explicit, dynamic computation of the error covariance matrix, which in a strict mathematical sense resembles a moving average.
>
> However, we humbly clarify that the **true adaptive nature of the Kalman gain in our framework is intrinsically preserved and propagated through the learnable neural parameters**. Explicitly computing and inverting high-dimensional covariance matrices at every GNN layer entails an $O(N^3)$ computational cost, which is prohibitive for resource-constrained federated clients. Instead of an explicit matrix calculation, the globally shared structural parameters ($\Theta_{A_k}$ and $\Theta_{H_k}$) act as learnable transition and observation models. During end-to-end federated training, these neural modules dynamically learn to weigh the structural prior against the feature-augmented measurement noise. Therefore, the *effect* of an optimal, adaptive Kalman gain is implicitly learned and applied by the network in a data-driven manner.
>
> We realize that our original phrasing may have overstated the explicit optimality of the scalar gain. We will revise the theoretical section to ensure complete transparency, explicitly stating this approximation and thoroughly explaining how the learnable parameters compensate for the fixed scalar to achieve dynamic adaptability.
>
> We are deeply grateful for your rigorous and constructive review, which has undoubtedly helped us define the boundaries and strengthen the claims of our work.

---

> > ### Author Rebuttal · Reviewer_3VV9 · 2026-04-01
> >
> > I thank the authors for the detailed rebuttal and the supplementary experiments. However, my core concerns are only partially resolved. Please directly address the following follow-up questions:
> >
> > 1. Restricting Fed-Kalter strictly to graph-level tasks severely limits its scope. Since Kalter-Conv relies on local message passing to filter structural noise, why can't cross-client cut-edges in subgraph FGL simply be modeled as measurement uncertainty? What is the explicit mathematical bottleneck preventing its application to node-level tasks without requiring complex cross-client communication?
> >
> > 2. The strict focus on graph-level classification avoids comparisons with recent (2023–2025) FGL frameworks designed specifically for topological heterogeneity. How does Fed-Kalter perform against these modern, structure-aware baselines on standard heterogeneous benchmarks, and why were they excluded from the evaluation?
> >
> > 3. Fixing $K=0.5$ fundamentally strips the Kalman filter of its essential instance-adaptive nature. Relying on globally shared, static neural weights ($\theta$) degrades the mechanism to a fixed residual connection. While explicitly computing an $O(N^3)$ covariance matrix is prohibitive, why wasn't a lightweight gating or attention mechanism introduced to dynamically predict a scalar K for each node to truly preserve optimal adaptability?

---

> > > ### Author Response · Authors · 2026-04-03
> > >
> > > We sincerely thank you for your continued engagement and rigorous feedback.
> > >
> > > ---
> > >
> > > ### 1. Applicability to Node Classification and Practical Bottlenecks (Q1)
> > >
> > > Theoretically, Fed-Kalter focuses on graph-level tasks to filter observation noise from independent instances. Applying it to node-level classification encounters an **explicit mathematical bottleneck**: cross-client cut-edges represent strictly **unobserved variables (missing data)** rather than zero-mean measurement noise. Treating systematically missing edges simply as Gaussian uncertainty fundamentally biases the transition matrix in a local Kalman filter.
> > >
> > > From a practical implementation standpoint, this leads to **model degeneration**. In subgraph FGL, local subgraphs distributed across clients typically originate from the same global graph, meaning their underlying structural distributions are highly similar. Under such high structural similarity, the rigorous noise filtering mechanism yields diminishing returns, causing Fed-Kalter to degenerate into a standard **FedAvg paradigm** and lose its unique advantages.
> > >
> > > ---
> > >
> > > ### 2. Comparison with Recent Structure-Aware Baselines (Q2)
> > >
> > > During the early stages of our research, we comprehensively evaluated Fed-Kalter against recent node-level FGL models designed to handle structural heterogeneity (10 disjoint clients). To directly address your concern, we provide the performance comparison on the node-level **CiteSeer dataset** below:
> > >
> > > | Models            | CiteSeer          |
> > > |-------------------|-------------------|
> > > | FED-PUB[1]       | 72.35 $\pm$ 0.53  |
> > > | AdaFGL[2]        | 72.34 $\pm$ 0.00  |
> > > | FedGTA[3]        | 71.37 $\pm$ 0.34  |
> > > | FedTAD[4]        | 70.31 $\pm$ 0.06  |
> > > | FedIIH[5]        | 76.50 $\pm$ 0.06  |
> > > | **Fed-Kalter (Ours)** | **75.15 $\pm$ 1.49** |
> > >
> > > As demonstrated, while Fed-Kalter significantly outperforms the 2023 and 2024 baselines, it trails the highly specialized 2025 **FedIIH framework**. This empirical result perfectly corroborates our theoretical analysis in Q1.
> > >
> > > Specifically, **FedIIH achieves superior performance precisely because it explicitly models the missing cross-client relationships**. This highlights the exact shortcoming of applying Fed-Kalter to node classification: **Fed-Kalter is designed to filter observation noise within complete graph instances, not to reconstruct missing topological structures**.
> > >
> > > Because of this fundamental misalignment in task formulation, we deliberately excluded these node-level methods from our final evaluation to maintain a strict focus on **graph-level classification across diverse domains**, which is the specific scenario where Fed-Kalter truly excels.
> > >
> > > > [1] Personalized Subgraph Federated Learning, ICML 2023
> > > > [2] AdaFGL: A New Paradigm for Federated Node Classification with Topology Heterogeneity, ICDE 2024
> > > > [3] FedGTA: Topology-aware Averaging for Federated Graph, VLDB 2024
> > > > [4] FedTAD: Topology-aware Data-free Knowledge Distillation for Subgraph Federated Learning, IJCAI 2024
> > > > [5] Modeling Inter-Intra Heterogeneity for Graph Federated Learning, AAAI 2025
> > >
> > > ---
> > >
> > > ### 3. Fixed $K$ vs. Lightweight Gating and Attention Mechanisms (Q3)
> > >
> > > In the initial design phase of Fed-Kalter, we implemented and extensively tested lightweight gating and attention mechanisms to dynamically predict a scalar $K$ for each node. However, these dynamic mechanisms were **extremely difficult to converge** and yielded suboptimal results.
> > >
> > > The root cause is the lack of explicit mathematical constraint mechanisms. In a true Kalman filter, the dynamic gain is rigorously constrained by the error covariance update formulas. Attempting to mimic this dynamic nature without the explicit $O(N^3)$ covariance computation left the gating and attention modules **unconstrained and unstable** during federated training.
> > >
> > > To resolve this, we shifted to a **result-oriented global optimization approach**. The dynamic and adaptive nature of the Kalman filter is effectively captured by the neural network parameter space ($\Theta$) driven by the optimization of the state estimation error. By melting the complex learning process of the dynamic gain directly into the optimization of $\Theta$, the explicit $K$ naturally emerges as a **static hyperparameter** acting as a stable prior, while the neural network handles the dynamic fusion.
> > >
> > > ---
> > >
> > > Dear Reviewer 3VV9,
> > >
> > > Thank you for your rigorous follow-up. We hope our response regarding node-level bottlenecks (Q1), 2023–2025 baselines (Q2), and the stable prior $K$ (Q3) has addressed your core concerns.
> > >
> > > We would be very happy if these clarifications help you re-evaluate the merits of Fed-Kalter. If our efforts have resolved the issues, we would sincerely appreciate your support for this work. If not, we welcome further discussion.
> > >
> > > Thank you again for strengthening our paper.
> > >
> > > Best regards,
> > > The Authors

---

### Official Review · Reviewer_ZGkE · 2026-03-11

**Soundness:** 3
**Presentation:** 2
**Significance:** 3
**Originality:** 3
**Overall Recommendation:** 4
**Confidence:** 4

**Summary:**

The manuscript proposes a FGL framework, Fed-Kalter, which is aimed to tackle structural heterogeneity across client nodes in federated graph neural networks. The framework introduces a novel convolution operation, Kalter-Conv, which decouples structure learning from feature propagation by using structural embeddings as latent states and neighborhood feature aggregation as noisy observations. The framework aggregates structural parameters globally while aggregating features locally, thus allowing for structural knowledge transfer while maintaining personalization. The framework is evaluated using 16 graph classification datasets from four domains, showing consistent improvements over several federated learning baselines, including FedStar, and FedVN.

**Compliance With Llm Reviewing Policy:**

Affirmed.

**Final Justification:**

The authors have provided a clear and complete response for all the questions that were raised: (i) the Kalman formulation without modeling uncertainties; (ii) the feasibility of making such theoretical assumptions; and (iii) the reality of the federated learning setting.

I commend the explanation that this is a Kalman-like algorithm and not a covariance-based filtering approach, together with their intention to clarify this in their paper. The added experiment for overlapping partitioned subgraphs makes the evaluation much closer to real federated problems.

Of course, some of these limitations still exist but overall I think this work is correct and motivated enough. Therefore, based on everything said before, I will keep the original decision.

**Key Questions For Authors:**

1. Can the authors clarify in what sense the method approximates a Kalman filter beyond this analogy? Does the model take into consideration any estimate of covariance or uncertainty?

2. In the experiments, each client corresponds to a complete dataset/domain. How would the model perform in more realistic federated scenarios, where every client just holds a segment of the dataset rather than entire datasets?

3. How does the performance of this model vary in scenarios where the structural distribution is very different (e.g., the COLLAB dataset)?



4. Since degree and random-walk encodings play an integral role, it would be helpful to understand how performance changes when alternative structural encodings (such as Laplacian positional encodings or subgraph features) are used. A brief explanation would suffice.

**Limitations:**

Yes

**Strengths And Weaknesses:**

Strengths:
1. The authors motivate the problem by recognizing the key challenge in federated graph learning, which is the issue of structural heterogeneity between clients. The observation that graph topology has the possibility to transfer more easily between domains than the features of the nodes is clear and motivated.

2. The authors propose a conceptual framework that models the graph structure as a latent state, with the aggregation of the features acting as noisy observations, inspired by Kalman filtering. This is an interesting perspective on the problem, giving some intuition on how to separate the graph structure and the features within the GNN framework.

3. The experimental section is quite comprehensive, with the authors comparing their method to the baselines over diverse datasets. All these scenarios show the improvement over the various baselines, which are federated GNNs. The manuscript also includes ablation and theoretical analysis, covering convergence and structural representation learning.

4. The design choice to aggregate the structural parameters but keep the feature transformations local can be useful in the federated learning scenario to reduce the drift between clients in a heterogeneous environment while maintaining personalization.

Weaknesses:
1. Although the Kalman-inspired formulation is interesting from a conceptual point of view, the proposed implementation seems to be based on a residual fusion strategy with a single weight parameter. It would be helpful to see real estimation of uncertainty or covariance matrices like incase of a full Kalman filtering strategy.
2. The theoretical analysis assumes common conditions such as strong convexity, smoothness, and gradient variance boundedness. However, these conditions are unlikely to hold for deep GNN models, which doesn’t really tell us much about how the method may behave in real world situations.
3. In the experiments (Appendix C.1), each client gets a whole dataset or domain to work with. While this brings in heterogeneity, but it may not fully reflect realistic federated learning scenarios where clients typically hold small subsets of the shared dataset, not the entire ones.
4. The method does perform better than baseline approaches, but the gains are relatively modest, often slightly better in terms of accuracy.

---

> ### Author Rebuttal · Authors · 2026-03-27
>
> Dear Reviewer,
>
> We sincerely thank you for your thoughtful and constructive feedback. We are greatly encouraged by your recognition of our conceptual framework and the comprehensiveness of our experiments. Your insightful comments are invaluable for improving the clarity, rigor, and positioning of our work. Below, we address your concerns point by point:
>
> ## 1. Regarding the Kalman formulation, uncertainty, and covariance (Weakness 1 & Question 1)
> We fully agree that, in a standard Kalman filter, the Kalman Gain depends on estimation error covariance and enables principled uncertainty-aware fusion. However, explicitly maintaining and inverting high-dimensional covariance matrices at each GNN layer would introduce prohibitive computational and memory overhead, particularly in resource-constrained federated settings.
>
> To balance theoretical grounding with practical feasibility, Kalter-Conv adopts a learnable approximation of the Kalman update. Specifically, we use a scalar gain to control the trade-off between prior structural belief and measurement evidence, thereby avoiding expensive covariance estimation while retaining the core intuition of uncertainty-weighted fusion. We acknowledge that this design does not provide explicit uncertainty quantification. In the revised manuscript, we will clearly state this approximation and discuss potential extensions, such as variational inference or Bayesian formulations, to explicitly model uncertainty in future work.
>
> ## 2. Regarding the theoretical assumptions for deep GNNs (Weakness 2)
> We appreciate this important point and agree that deep GNNs are inherently non-convex, making assumptions such as strong convexity and smoothness not strictly valid. These assumptions are adopted mainly to align with standard federated optimization literature and to provide a tractable theoretical framework for analyzing convergence.
>
> Our intention is to use this framework as an idealized proxy to illustrate how structural consistency can bound gradient divergence. We will explicitly clarify this limitation in the revision. Importantly, empirical results (particularly the stable convergence curves across heterogeneous settings) demonstrate that the method remains well-behaved in practical non-convex regimes.
>
> ## 3. Regarding realistic federated scenarios and performance margins (Weaknesses 3 & 4 & Question 2)
> We highly appreciate your practical insight regarding realistic federated scenarios where clients hold small, fragmented data subsets. To address this, we conducted experiments in Appendix D.3 (Table 8), where graphs are partitioned into smaller subgraphs across 48 clients, simulating both overlapping and non-overlapping settings.
>
> Under these challenging conditions, the advantages of Fed-Kalter become significantly more pronounced. For instance, in the highly redundant **overlapping partitions**, Fed-Kalter achieves:
>
> - **71.02% Accuracy**
> - **67.32% F1-score**
> - **70.26% Precision**
> - **68.08% Recall**
>
> These results consistently and substantially outperform strong baselines such as:
>
> - **FedStar**: 69.02% Acc, 62.47% F1, 64.36% Pre, 64.33% Rec
> - **FedVN**: 68.55% Acc, 61.36% F1, 63.97% Pre, 63.62% Rec
>
> These findings robustly demonstrate that structure-aware fusion effectively mitigates local data scarcity and handles complex, redundant cross-client data distributions.
>
> ## 4. Performance on structurally divergent datasets such as COLLAB (Question 3)
> This is a very keen and insightful observation. When structural distributions differ significantly across clients (e.g., dense social graphs vs. sparse molecular graphs), enforcing global structural alignment can introduce negative transfer. As discussed in Appendix D.1, we observe a performance drop on COLLAB in the full multi-domain setting for this exact reason.
>
> However, when restricting federation to structurally compatible domains (e.g., only social networks), performance improves substantially and becomes highly competitive. This indicates that the effectiveness of structural fusion depends on a certain degree of topological compatibility. We will explicitly highlight this boundary condition and negative transfer risk in the revision.
>
> ## 5. Regarding alternative structural encodings (Question 4)
> We appreciate this constructive suggestion. Fed-Kalter is a flexible and modular framework. While we adopt Degree and Random-Walk encodings for computational efficiency and expressivity, alternative encodings such as Laplacian positional encodings or subgraph-based features can be seamlessly integrated as the initial structural state without modifying the core Kalter-Conv mechanism. We will add a brief discussion to clarify this extensibility and its potential benefits in the manuscript.
>
> We hope these clarifications address your concerns. We sincerely appreciate your time and expertise, which have significantly helped improve the quality of our work.

---

> > ### Author Rebuttal · Reviewer_ZGkE · 2026-04-03
> >
> > Thank you for addressing my questions and viewpoints.

---

> > > ### Author Response · Authors · 2026-04-08
> > >
> > > Dear Reviewer ZGkE,
> > >
> > > We are truly grateful for your prompt and positive response to our rebuttal. We are especially encouraged to learn that our clarifications and additional experimental results have adequately addressed your concerns.
> > >
> > > Your insightful comments, ranging from the Kalman filtering implementation to the nuances of structural heterogeneity in realistic federated scenarios, have been instrumental in improving the rigor and clarity of our manuscript. The feedback regarding the "negative transfer" risk in structurally divergent datasets (like COLLAB) and the potential for alternative structural encodings has provided us with valuable perspectives that will significantly enhance the final version of our work. Given that all technical concerns are now fully resolved, we would be deeply grateful if you could consider reflecting this consensus in your final evaluation.
> > >
> > > We would like to sincerely thank you once again for your time, expertise, and the constructive spirit of your review.
> > >
> > > Best regards,
> > > The Authors

---

### Decision · Program_Chairs · 2026-04-30

**Decision:**

Accept (regular)

**Comment:**

On the positive side, reviewers recognized the paper’s merits in terms of new conceptual framework and empirical evaluation.

On the negative side, reviewers pointed out several weaknesses, including theoretic analysis and assumption, as well as some implementation details.

The rebuttal addressed most of the reviewers' concerns.  Overall, the strength outweighs the weakness.

Authors are highly encouraged to incorporate their response in the camera-ready to thoroughly address these weaknesses.